# 🤖 PediatricsGPT: Large Language Models as Chinese Medical Assistants for Pediatric Applications

**Dingkang Yang**[1,3][8][†]    **Jinjie Wei**[1,3][†]    **Dongling Xiao**[2][†]    **Shunli Wang**[1][§]    **Tong Wu**[2][§]
**Gang Li**[2][§]    **Mingcheng Li**[1][§]    **Shuaibing Wang**[1][§]    **Jiawei Chen**[1][§]    **Yue Jiang**[1][§]
**Qingyao Xu**[1][§]    **Ke Li**[2][§]    **Peng Zhai**[1,3][*]    **Lihua Zhang**[1,3,4,5][*]

[1] Academy for Engineering and Technology, Fudan University, Shanghai, China
[2] Tencent Youtu Lab, Shanghai, China
[3] Cognition and Intelligent Technology Laboratory, Shanghai, China
[4] Engineering Research Center of AI and Robotics, Ministry of Education, Shanghai, China
[5] AI and Unmanned Systems Engineering Research Center of Jilin Province, Changchun, China
{dkyang20, pzhai, lihuazhang}@fudan.edu.cn
jjwei23@m.fudan.edu.cn, xdluestc@outlook.com
tristanli@tencent.com

## Abstract

Developing intelligent pediatric consultation systems offers promising prospects for improving diagnostic efficiency, especially in China, where healthcare resources are scarce. Despite recent advances in Large Language Models (LLMs) for Chinese medicine, their performance is sub-optimal in pediatric applications due to inadequate instruction data and vulnerable training procedures. To address the above issues, this paper builds PedCorpus, a high-quality dataset of over 300,000 multi-task instructions from pediatric textbooks, guidelines, and knowledge graph resources to fulfil diverse diagnostic demands. Upon well-designed PedCorpus, we propose PediatricsGPT, the first Chinese pediatric LLM assistant built on a systematic and robust training pipeline. In the continuous pre-training phase, we introduce a hybrid instruction pre-training mechanism to mitigate the internal-injected knowledge inconsistency of LLMs for medical domain adaptation. Immediately, the full-parameter Supervised Fine-Tuning (SFT) is utilized to incorporate the general medical knowledge schema into the models. After that, we devise a direct following preference optimization to enhance the generation of pediatrician-like humanistic responses. In the parameter-efficient secondary SFT phase, a mixture of universal-specific experts strategy is presented to resolve the competency conflict between medical generalist and pediatric expertise mastery. Extensive results based on the metrics, GPT-4, and doctor evaluations on distinct downstream tasks show that PediatricsGPT consistently outperforms previous Chinese medical LLMs. The project and data will be released at https://github.com/ydk122024/PediatricsGPT.

## 1 Introduction

As an essential component of medicine, pediatrics plays an indispensable role in ensuring children's health growth [22, 23]. The unbalanced distribution of healthcare resources [36] has resulted in a massive shortage of pediatricians, especially in populous countries led by China [37, 19]. With the

---

[†]Equal first contributions. [§]Equal second contributions. [*]Corresponding authors. [8]Project lead.

38th Conference on Neural Information Processing Systems (NeurIPS 2024).

rapid advances in LLMs exemplified by ChatGPT [33], developing intelligent pediatric consultation systems provides promise for enriching medical services. Although Chinese LLMs [18, 59, 2, 57, 20] have exhibited progress in general language understanding, they are incompetent in the pediatric medical field due to the lack of domain-specific discipline and specialized expertise injection.

To fulfil the interactive demands of Chinese medicine, preliminary efforts [8, 45, 50, 15] have enhanced LLMs' healthcare mastery through Supervised Fine-Tuning (SFT) training and medically relevant corpus collection. Despite improvements, challenges remain due to unavoidable dilemmas, including inadequate instruction data and vulnerable training procedures. Specifically, (i) existing instruction data typically involve vanilla rephrasing of the general medical corpus [50] or aggregation of doctor-like dialogues [56], which loses the specialization and focus in pediatric applications. More importantly, the current straightforward different round instruction construction paradigms [58, 15] fail to accommodate multi-task healthcare services in real-world scenarios, limiting the model generalization and inducing response hallucination. (ii) Furthermore, prior methods mostly relied on SFT to compensate for medical instruction following capabilities, ignoring the discrepancies between inherent and externally absorbed knowledge within the models. This single pattern causes secondary LLMs to lapse into excessive role-playing rather than understanding [40]. Despite a few attempts in the pre-training and Reinforcement Learning from Human Feedback (RLHF) phases [7, 34], their performance is restricted by actor-critic instability [41] and online sampling bias [61].

Motivated by these observations, we construct PedCorpus, a high-quality dataset with over 300,000 instructions across single-turn and multi-turn medical conversations. Besides containing generalist healthcare data, PedCorpus incorporates multi-dimensional corpora from pediatric textbooks, guidelines, and knowledge graphs to ensure medical knowledge's accuracy. Vanilla instructions can also be readily extended to seed instructions for generating specialized corpora to serve different training phases. Furthermore, we integrate the well-presented GPT-4-distilled data with authentic doctor-patient dialogue data to standardize the fluency and faithfulness of instruction information.

Among our PedCorpus, we propose PediatricsGPT, the first Chinese pediatric LLM assistant with pediatric expertise and medical generalist. PediatricsGPT is developed on a systematic training pipeline that includes Continuous Pre-Training (CPT), full-parameter SFT, human preference alignment, and parameter-efficient secondary SFT. In this case, we introduce a hybrid instruction pre-training mechanism in CPT to bridge the capability weakening due to corpus format discrepancies between the internal and injected medical knowledge of foundation models, facilitating knowledge accumulation and extension. Meanwhile, a Direct Following Preference Optimization (DFPO) in human preference alignment is devised to enhance response robustness and align human preferences. Additionally, we present a mixture of universal-specific experts strategy to tackle the competency conflict between medical generalist and pediatric expertise in secondary SFT via Low-Rank Adaptation (LoRA) [27], which strengthens the model's adaptability to distinct downstream tasks. We conduct three pragmatic pediatric tasks to evaluate the different capabilities of existing models. Extensive experiments on pediatric and public benchmarks show that our PediatricsGPT outperforms open-source Chinese medical LLMs and baselines, yielding competitive performance compared to GPT-3.5-turbo.

## 2 Related Work

**Chinese Large Language Model Evolution.** The emergence of Large Language Models (LLMs) dominated by ChatGPT [33] and GPT-4 [5] has revolutionized the paradigm for novel human-machine interaction. Driven by learning-oriented technologies [11–13, 53–55, 48], pragmatic instruction [32, 47] and preference optimization [7, 34] strategies enable LLMs to address complex generation tasks with aligned human intentions. Despite improvements, large-scale resources for training general LLMs [28, 43, 44] are anchored in the English corpora, limiting their abilities to respond reliably in extensive Chinese application scenarios. Recently, researchers [18, 59] have attempted to enhance the comprehension and execution of Chinese instructions in open-source LLMs by augmenting Chinese vocabulary and data (*e.g.,* Chinese LLaMA and Alpaca [18]). To facilitate Chinese-specific demands, several LLMs trained from scratch exhibit remarkable Chinese proficiency due to multilingual data resources, such as the Baichuan [2, 52], General Language Model (GLM) [20, 57], and Qwen [6] families. In this work, the Baichuan2-Base series is utilized as the foundation model for our PediatricsGPT, given its comprehensive potential among similar contenders.

**LLMs in Medical Applications.** Current LLMs provide unprecedented opportunities to develop resource-efficient and diagnostic-comprehensive intelligent healthcare systems. Despite universal

Table 1: Statistical information on the proposed dataset. PedCorpus is well extensible and adaptable by incorporating general domain data and as seed instructions to generate specialized corpora (*i.e.,* PedCorpus-CPT and PedCorpus-DFPO). "KG" means the Knowledge Graphs.

| Dataset | Data Sources | Department | Number/Size | Human Preference | Task Type | | |
| --- | --- | --- | --- | --- | --- | --- | --- |
| | | | | | MedKQ&A | EviDiag | TreRecom |
| PedCorpus | Pediatric Textbooks | Pediatrics | 37,284 | ✔ | ✔ | – | ✔ |
| | Pediatric Guidelines | Pediatrics | 63,129 | ✔ | ✔ | – | ✔ |
| | Pediatric KG | Pediatrics | 46,320 | ✔ | ✔ | – | ✔ |
| | Real Doctor-Patient Conversations | Multiple | 46,385 | ✔ | – | ✔ | ✔ |
| | Distilled Medical Datasets | Multiple | 107,177 | – | ✔ | ✔ | ✔ |
| PedCorpus-CPT | Plain Textbooks, Guidelines, KG | Multiple | | – | ✔ | ✔ | ✔ |
| | Filtered Chinese Wikipedia | Multiple | 975.8MB | – | – | – | – |
| | Extended data from PedCorpus | Multiple | | – | – | – | – |
| PedCorpus-DFPO | Pediatrics data from PedCorpus | Pediatrics | 15,556 | ✔ | ✔ | ✔ | ✔ |

models [5, 33] equipped with certain internal knowledge regarding biomedicine, they are incompetent in real-world medical applications due to the absence of domain-specific disciplines. In this context, several efforts [45, 50, 15, 30] attempt to construct medically tailored LLMs from multiple perspectives. For instance, ChatDoctor [30] uses patient-doctor conversation data based on LLaMA [43] to enhance the language model's accuracy in healthcare. DoctorGLM [50] proves that a healthcare-purpose LLM can be implemented with affordable overhead by fine-tuning ChatGLM-6B [20]. After that, more Chinese medical LLMs [51, 8, 58, 14, 56] are progressively presented to generate doctor-like robust responses, such as HuatuoGPT [58], DISC-MedLLM [8], and Zhongjing [56]. Despite advances in general medical knowledge, current models are suboptimal for pressing pediatric applications. In comparison, our sophisticated training procedure and high-quality instruction datasets inject new insights and prospects for developing specialized LLMs with pediatric expertise.

## 3 Methodology

This section describes the proposed PedCorpus dataset and the sequential pipeline for developing PediatricsGPT. Figure 1 illustrates the comprehensive method workflow.

### 3.1 PedCorpus: Multi-task Medical Instruction Dataset

To endow the model with versatile diagnostic proficiency, PedCorpus is constructed through the multi-dimensional corpus across three application-oriented medical tasks, including Knowledge Question-Answer (MedKQ&A), Evidence-based Diagnosis (EviDiag), and Treatment Recommendation (TreRecom). Table 1 shows the detailed statistical information from different data sources. We explain the three patterns of PedCorpus construction below.

**Specialized Pediatric Data.** Extracting pediatric data from textbooks, guidelines, and knowledge graphs ensures knowledge professionalism. Specifically, we automatically extract standard medical definitions and descriptions from physical textbooks covering 131 disease types in 11 broad categories. Over 500 corresponding disease guidelines are collected, including diagnostic protocols and treatment consensus. Additionally, extensive knowledge entities are sampled from ternary instances in the knowledge graphs. Based on these resources, we introduce a role-playing-driven instruction building rule via GPT-4 API that produces well-organized instructions to enable **accurate** and **humanistic** model responses. The detailed building procedure is shown in Appendix A.1.

**Real Doctor-patient Conversations.** To avoid the model collapse dilemma [42], we incorporate authentic doctor-patient dialogues from online treatment platforms and voice transcriptions during medical consultations. The single-/multi-turn instructions are jointly considered to equip the model with healthcare interrogation and contextual understanding. Original responses from real doctors are usually terse and noisy, potentially worsening the generation quality [58]. To this end, we craft 100 high-quality examples to guide the advanced language model by the in-context learning to regularize vanilla conversations in the self-instruct pattern [17, 46]. This approach ensures **doctor-like** and **patient-friendly** model responses. More regularization details are shown in Appendix A.2.

**Distilled Medical Datasets.** Integrating general medical knowledge from existing datasets [29, 26, 60] is a common practice in previous efforts [15, 50, 51, 8]. However, we find numerous unclear and incomplete representations in the instruction instances from public benchmarks due to the absence of careful calibration, potentially triggering hallucinated outputs. Consequently, we manually sample

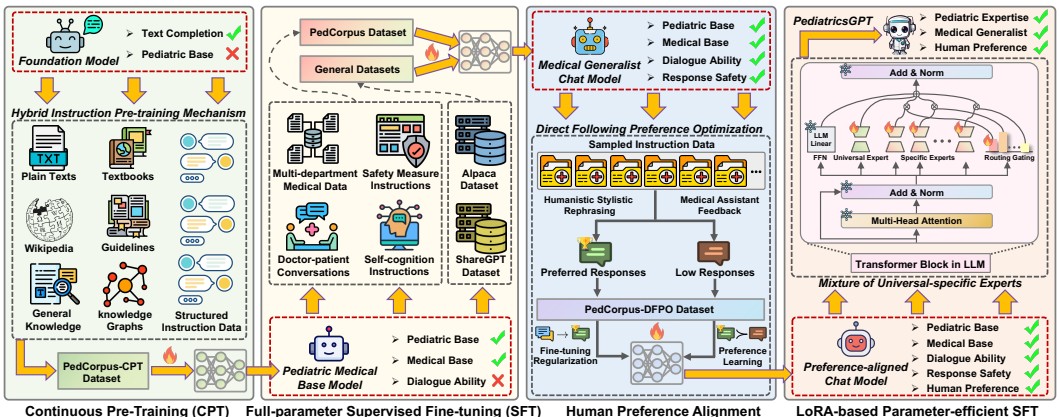

Figure 1: The sequential pipeline for developing PediatricsGPT. We begin by injecting intensive medical and world knowledge into the foundation model through the hybrid instruction mechanism in CPT phase. Then, full-parameter SFT is implemented to improve the model's instruction-following capabilities regarding medical generalists. After that, we introduce the direct following preference optimization to control the model behaviour to align with human preference. In the parameter-efficient SFT phase, the LoRA-based mixture of universal-specific experts is devised to mitigate conflicts across downstream tasks and competition between pediatric expertise and general mastery.

107,177 knowledge-intensive instructions from three mainstream benchmarks (*i.e.,* Huatuo-26M [29], MedDialog [26], and CMeKG [10]), adhering to the philosophy of quality over quantity [62]. After that, a progressive instruction reconstruction rule is proposed to distill the sampled instructions to ensure **informative** and **logical** model responses. The rule process can be found in Appendix A.3.

## 3.2   Hybrid Instruction Pre-training in CPT

Continuous Pre-Training (CPT) is essential in developing domain-specific models [14, 49, 56] since it can break the scaling law [24] to a certain extent. For this purpose, we introduce the PedCorpus-CPT dataset to ensure a high-quality pre-training corpus. From Table 1, PedCorpus-CPT consists of three-part data components. (i) We integrate plain texts from vanilla pediatric textbooks, guidelines, and knowledge graphs. (ii) The filtered Chinese Wikipedia [3] is also considered to achieve the model's trade-off for medical-general knowledge memory capacity. (iii) In practice, we observe that CPT leads to catastrophic forgetting of the models at follow-up due to different data distribution and format discrepancies compared to the original pre-training and SFT. Thus, we introduce a hybrid instruction pre-training mechanism to bridge these discrepancies. The core philosophy is to assemble instruction data from PedCorpus with `Input-Output` forms into `Completion` forms, which are then assimilated into plain texts to provide multi-task and complementary information. This mechanism effectively mitigates inconsistencies between the internal-injected medical knowledge of the foundation model while reinforcing medical domain adaptation. Moreover, we take PedCorpus as the seed instructions to improve multiple-department corpus density and breadth via knowledge-enhanced prompts. The prompt template is shown in Appendix B.

We pre-train the foundation model to follow the causal language modelling paradigm. Given any input token sequence $\boldsymbol{t} = (t_0, t_1, t_2, ...) \in \mathcal{D}_{cpt}$ from the above multi-channel corpus $\mathcal{D}_{cpt}$, the next token $t_i$ is autoregressively predicted by minimizing the negative log-likelihood:

$$\mathcal{L}_{\text{CPT}}(\theta, \mathcal{D}_{cpt}) = \mathbb{E}_{\boldsymbol{t} \sim \mathcal{D}_{cpt}} \left[ -\sum_i^{|\boldsymbol{t}|} \log p(t_i \mid t_0, t_1, ..., t_{i-1}; \theta) \right], \tag{1}$$

where $\theta$ is the model parameter and the input context consists of $t_0, t_1, ..., t_{i-1}$.

## 3.3   Full-parameter Supervised Fine-tuning

During this phase, we activate the model's ability to follow medical instructions by the Full-parameter Supervised Fine-tuning (FSFT). The full-parameter pattern enables a fuller invocation of the intensive knowledge in CPT and promotes comprehension and logical reasoning about diverse structured instructions. The training data at this phase is composed of the following three aspects. (i) We utilize the multi-department medical data in the PedCorpus dataset to develop the medical generalist. (ii)

Chinese instruction data (*i.e.,* Alpaca dataset [35] and ShareGPT [4]) from general domains are selectively integrated to avoid the potential overfitting risk. (iii) Providing safety measures is vital for LLM assistants yet overlooked by prior methods [62]. In contrast, we write 200 training instructions with some degree of maliciousness, hallucinations, and counterfactuals. Correspondingly, the refusal responses with detailed explanations for disobedience are carefully crafted. We also include 300 examples related to self-cognition content. These data significantly improve the robustness and security of the model against unfriendly commands.

Given any input instruction $\boldsymbol{x} = (x_0, x_1, x_2, ...) \in \mathcal{D}_{fsft}$ and corresponding target response $\boldsymbol{y} = (y_0, y_1, y_2, ...) \in \mathcal{D}_{fsft}$ from the above-integrated fine-tuning dataset $\mathcal{D}_{fsft}$, the optimization objective can be formulated as follows:

$$\mathcal{L}_{\text{FSFT}}(\theta, \mathcal{D}_{fsft}) = \mathbb{E}_{(\boldsymbol{x},\boldsymbol{y})\sim\mathcal{D}_{fsft}} \left[ -\textstyle\sum_{i=1}^{|\boldsymbol{y}|} \log p(y_i \mid \boldsymbol{x}, y_{<i}; \theta) \right]. \tag{2}$$

### 3.4 Direct Following Preference Optimization

Aligning human intention preferences facilitates the model to generate harmless responses. To this end, we introduce PedCorpus-DFPO $\mathcal{D}_{dfpo}$, a preference dataset to guide the model in learning human preference behaviours. PedCorpus-DFPO contains the input instruction set $\boldsymbol{x} = (x_0, x_1, x_2, ...) \in \mathcal{D}_{dfpo}$, which is selectively sampled from vanilla PedCorpus. On the one hand, we perform a humanistic stylistic rephrasing of the outputs to generate preferred responses $\boldsymbol{y}^w = (y_0^w, y_1^w, y_2^w, ...) \in \mathcal{D}_{dfpo}$. On the other hand, the corresponding low responses $\boldsymbol{y}^l = (y_0^l, y_1^l, y_2^l, ...) \in \mathcal{D}_{dfpo}$ are generated from the feedback of a low-capability medical assistant [45] to maintain domain consistency.

Despite impressive improvements achieved by RLHF-based approaches [56, 58], challenges remain due to unstable reward modelling and significant computational costs [41, 61]. Inspired by single-stage preference learning [38], we propose a stable and lightweight method for domain-specific LLMs called Direct Following Preference Optimization (DFPO). DFPO utilizes variable changes to formulate the preference loss as a policy function that efficiently optimizes the policy with a simple binary cross-entropy objective. Meanwhile, our method directly regularizes model behaviour boundaries in an instruction-following paradigm on medical demonstrations of preferred responses, facilitating robustness and smoothing of the preference learning.

Theoretically, the observed probability of a particular preference pair usually follows the Bradley-Terry model [9], and $\boldsymbol{y}^w$ is preferred over $\boldsymbol{y}^l$ (denoted $\boldsymbol{y}^w \succ \boldsymbol{y}^l$):

$$p(\boldsymbol{y}^w \succ \boldsymbol{y}^l) = \sigma(\gamma(\boldsymbol{x}, \boldsymbol{y}^w) - \gamma(\boldsymbol{x}, \boldsymbol{y}^l)), \tag{3}$$

where $\gamma(\boldsymbol{x}, \boldsymbol{y}^{w/l})$ means the parameterized reward function and $\sigma(\cdot)$ is the sigmoid activation. In this case, the overall optimization objective is expressed as:

$$\mathcal{L}_{\text{DFPO}}(\theta, \mathcal{D}_{dfpo}) = -\mathbb{E}_{(\boldsymbol{x},\boldsymbol{y}^w,\boldsymbol{y}^l)\sim\mathcal{D}_{dfpo}} \left[ \log \sigma(\beta \log \frac{\pi_\theta(\boldsymbol{y}^w \mid \boldsymbol{x})}{\pi_r(\boldsymbol{y}^w \mid \boldsymbol{x})} - \beta \log \frac{\pi_\theta(\boldsymbol{y}^l \mid \boldsymbol{x})}{\pi_r(\boldsymbol{y}^l \mid \boldsymbol{x})}) \right] + \mu\Phi(\boldsymbol{x}, \boldsymbol{y}^w), \tag{4}$$

where $\pi_\theta$ and $\pi_r$ are the desired optimal policy and the reference policy, respectively. $\beta$ is the control parameter reflecting the deviation from the basic $\pi_r$. For the fine-tuning regularization term $\Phi(\boldsymbol{x}, \boldsymbol{y}^w)$ with the scaling coefficient $\mu$, the implementation process is equivalent to maximizing the log probability $p(\boldsymbol{y}^w \mid \boldsymbol{x})$ regarding the preferred responses $\boldsymbol{y}^w$ given the input instructions $\boldsymbol{x}$:

$$\Phi(\boldsymbol{x}, \boldsymbol{y}^w) = \mathbb{E}_{(\boldsymbol{x},\boldsymbol{y}^w)\sim\mathcal{D}_{dfpo}} \left[ -\textstyle\sum_{i=1}^{|\boldsymbol{y}^w|} \log p(y_i^w \mid \boldsymbol{x}, y_{<i}^w; \theta) \right]. \tag{5}$$

### 3.5 Mixture of Universal-specific Experts in Parameter-efficient SFT

This phase aims to reinforce the model performance for various pediatric applications through the LoRA-based Parameter-efficient SFT (PSFT). The used dataset $\mathcal{D}_{psft}$ is derived from the pediatric department in PedCorpus and partial general medical/world data. In practice, we observe that competition across different pediatric tasks and the conflicts between medical generalization and specialized knowledge deteriorate instruction-following abilities. Accordingly, we propose a mixture of universal-specific experts strategy to address these challenges. Formally, LoRA adapters [27] act as experts to replace the linear layers in the Feed-Forward Neural (FFN) networks of LLMs, providing

Table 2: Comparison results of different models on three pediatric medical benchmarks. In each benchmark, the best results are marked in **bold**, and the second-best results are marked underlined.

| Benchmark | Model | ROUGE-1 | ROUGE-2 | ROUGE-L | BLEU-1 | BLEU-2 | BLEU-3 | BLEU-4 | GLEU | Distinct-1 | Distinct-2 |
|---|---|---|---|---|---|---|---|---|---|---|---|
| MedKQ&A | Baichuan2-7B | 40.88 | 19.44 | 21.50 | 26.77 | 20.00 | 17.30 | 14.86 | 24.88 | 20.14 | 39.95 |
| | Baichuan2-13B | 46.96 | 22.85 | 22.54 | 29.02 | 25.62 | 22.63 | 19.31 | 27.97 | 21.45 | 42.53 |
| | HuatuoGPT | 48.52 | 23.44 | 25.13 | 43.00 | 41.25 | 36.31 | 29.82 | 34.60 | 20.42 | 41.27 |
| | DISC-MedLLM | 53.83 | 25.98 | 27.71 | 47.91 | 44.57 | 37.65 | 30.07 | 37.11 | 26.63 | 51.98 |
| | Zhongjing | 53.97 | 26.03 | 29.56 | 51.11 | 45.04 | 39.13 | 33.59 | 42.61 | **26.75** | **52.66** |
| | HuatuoGPT-II | 55.27 | 26.59 | 27.95 | 59.07 | 51.49 | 45.38 | 38.70 | 39.18 | 20.97 | 41.34 |
| | Meditron-7B | 55.63 | 26.19 | 30.37 | 58.43 | 53.45 | 56.07 | 38.77 | 42.23 | 22.34 | 45.17 |
| | Llama3.1-8B | 53.18 | 24.74 | 28.26 | 45.07 | 42.45 | 36.57 | 29.73 | 35.63 | 22.74 | 46.52 |
| | ChatGPT | 56.92 | 27.87 | 29.05 | 61.58 | 54.37 | 47.97 | 40.77 | 45.15 | 20.76 | 40.19 |
| | GPT-4 | 58.79 | 33.56 | 32.15 | **62.53** | 59.14 | 55.26 | 52.39 | 53.72 | 21.79 | 43.26 |
| | PediatricsGPT-7B | 58.08 | 31.78 | 31.11 | 59.41 | 56.88 | 57.47 | 55.34 | 54.41 | 24.33 | 47.41 |
| | PediatricsGPT-13B | **60.85** | **36.56** | **35.64** | 61.65 | **63.17** | **58.96** | **59.34** | **57.22** | 24.24 | 46.23 |
| EviDiag | Baichuan2-7B | 26.81 | 7.75 | 11.22 | 15.18 | 11.51 | 9.19 | 6.72 | 13.44 | 23.65 | 46.93 |
| | Baichuan2-13B | 39.14 | 12.06 | 12.44 | 47.65 | 36.02 | 28.82 | 21.19 | 28.28 | 25.45 | 50.43 |
| | HuatuoGPT | 35.12 | 10.77 | 15.04 | 46.22 | 33.10 | 25.44 | 21.22 | 25.44 | 22.30 | 45.73 |
| | DISC-MedLLM | 33.55 | 11.67 | 15.32 | 15.91 | 12.46 | 10.27 | 7.96 | 16.77 | **35.89** | **69.36** |
| | Zhongjing | 40.92 | 14.26 | 17.41 | 48.64 | 37.52 | 30.17 | 22.44 | 27.03 | 33.40 | 65.89 |
| | HuatuoGPT-II | 39.52 | 12.14 | 16.38 | 49.58 | 37.62 | 30.66 | 23.34 | 28.98 | 21.97 | 43.62 |
| | Meditron-7B | 42.63 | 15.12 | 18.94 | 52.36 | 39.24 | 37.78 | 27.15 | 31.25 | 22.07 | 45.13 |
| | Llama3.1-8B | 37.25 | 13.07 | 16.23 | 44.54 | 32.29 | 23.72 | 20.12 | 23.57 | 24.43 | 45.67 |
| | ChatGPT | 40.88 | 13.42 | 16.97 | 48.84 | 37.69 | 30.55 | 23.17 | 29.02 | 23.49 | 46.54 |
| | GPT-4 | **48.48** | 16.74 | 21.51 | 57.59 | 44.78 | 37.94 | 30.56 | 36.79 | 25.69 | 50.13 |
| | PediatricsGPT-7B | 45.83 | 16.60 | 19.91 | 54.37 | 41.99 | 37.59 | 29.03 | 33.42 | 23.49 | 46.61 |
| | PediatricsGPT-13B | 47.32 | **17.63** | **21.87** | **58.21** | **45.72** | **39.74** | **31.25** | **37.15** | 23.34 | 46.34 |
| TreRecom | Baichuan2-7B | 48.39 | 23.07 | 26.35 | 47.94 | 40.91 | 35.54 | 29.69 | 35.06 | 21.90 | 43.57 |
| | Baichuan2-13B | 48.87 | 23.41 | 26.42 | 49.96 | 46.24 | 42.84 | 35.04 | 35.63 | 22.36 | 45.12 |
| | HuatuoGPT | 53.48 | 25.41 | 27.08 | 58.14 | 49.64 | 42.93 | 35.16 | 41.63 | 23.26 | 46.21 |
| | DISC-MedLLM | 52.77 | 24.26 | 28.89 | 58.73 | 50.05 | 42.96 | 35.59 | 42.44 | 24.30 | 51.95 |
| | Zhongjing | 54.92 | 26.63 | 29.68 | 60.12 | 53.31 | 44.25 | 38.76 | 40.38 | 26.18 | 53.94 |
| | HuatuoGPT-II | 58.44 | 30.47 | 32.02 | 59.91 | 54.26 | 45.73 | 38.92 | 42.28 | 28.88 | 57.15 |
| | Meditron-7B | 58.56 | 32.25 | 33.37 | 60.47 | 55.36 | 48.73 | 42.18 | 46.73 | 28.51 | 57.45 |
| | Llama3.1-8B | 52.45 | 24.98 | 26.14 | 57.56 | 48.11 | 41.67 | 24.03 | 40.67 | 22.73 | 45.13 |
| | ChatGPT | 59.59 | 33.34 | 35.79 | 62.81 | 55.79 | 49.85 | 43.29 | 47.59 | 31.09 | 56.87 |
| | GPT-4 | 61.94 | 37.27 | 36.73 | 63.23 | 56.24 | 50.58 | 44.07 | **55.26** | 30.27 | 56.48 |
| | PediatricsGPT-7B | 56.92 | 29.13 | 31.26 | 61.36 | 55.34 | 46.44 | 40.61 | 44.65 | 26.06 | 52.77 |
| | PediatricsGPT-13B | **62.83** | **39.32** | **40.82** | **63.56** | **56.68** | **50.80** | **44.31** | 54.65 | **31.94** | **57.56** |

trainable parameters. Several specific experts $\left\{E_j^s\right\}_{j=1}^{T}$ are assigned adaptive activations to master distinct pediatric expertise through soft routing. The routing gating is defined as follows:

$$G(\boldsymbol{x}) = \text{Softmax}(\boldsymbol{x}\boldsymbol{W}_g + \mathcal{S}(\varphi(\boldsymbol{x}\boldsymbol{W}_n)). \tag{6}$$

$\boldsymbol{W}_g$ and $\boldsymbol{W}_n$ are the learnable weights. $\mathcal{S}(\varphi(\boldsymbol{x}\boldsymbol{W}_n)$ is the noise term for regularizing the expert utilization balance, where $\mathcal{S}(\cdot)$ and $\varphi(\cdot)$ represent the Standard Normal distribution sampling and Softplus function, respectively. Moreover, we consistently activate a universal expert $E^u$ across all training data to prevent general knowledge forgetting and mitigate competency conflict. The parameterized output $\boldsymbol{z}$ of all the experts in the forward process can be mathematized as follows:

$$\boldsymbol{z} = \frac{\alpha}{r}\left(\sum_{j=1}^{T} G(\boldsymbol{x})_j E_j^s(\boldsymbol{x}) + E^u(\boldsymbol{x})\right), \tag{7}$$

where $r$ is the rank value and $\alpha$ is a hyper-parameter for approximating the learning rate.

## 4 Experiments

### 4.1 Datasets and Implementation Details

Extensive experiments are conducted on three application-oriented benchmarks to assess the model's pediatric medical abilities, including Knowledge Question-Answer (**MedKQ&A**), Evidence-based Diagnosis (**EviDiag**), and Treatment Recommendation (**TreRecom**). Each benchmark contains 300 held-out samples to reject data leakage during training. In addition, we select two publicly available Chinese medical benchmarks to validate the model's generalizability in general healthcare. Specifically, we sample 50 challenging instances of diagnostic queries from each department from the **webMedQA** [25] and **CMD** [1] benchmarks, respectively, leading to testing sets with 300 samples.

Our PediatricsGPT is developed upon the Baichuan2-Base [52] models in two versions with 7 and 13 billion parameters. The model training is accomplished through the PyTorch platform with Accelerate

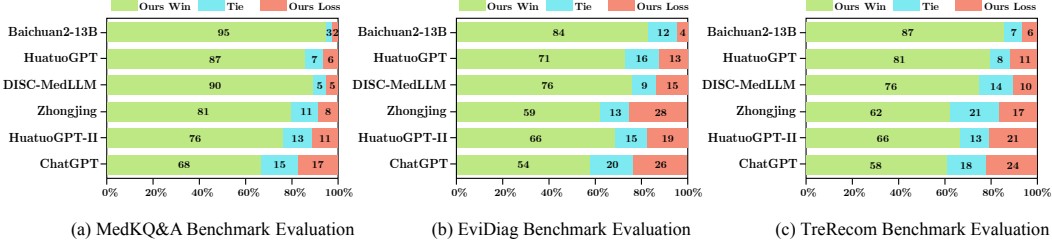

(a) MedKQ&A Benchmark Evaluation    (b) EviDiag Benchmark Evaluation    (c) TreRecom Benchmark Evaluation

Figure 2: Response comparisons of PediatricsGPT-13B with other baselines via GPT-4 evaluation.

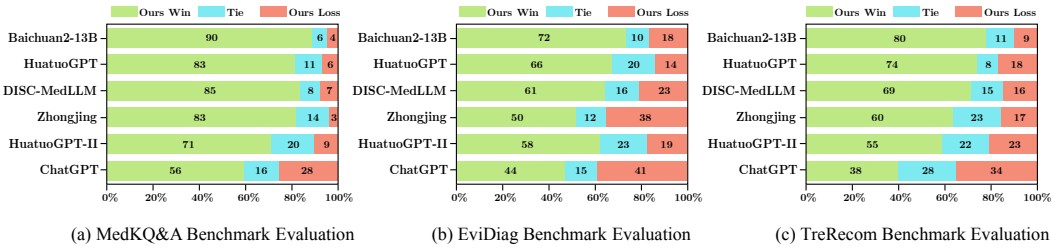

(a) MedKQ&A Benchmark Evaluation    (b) EviDiag Benchmark Evaluation    (c) TreRecom Benchmark Evaluation

Figure 3: Response comparisons of PediatricsGPT-13B with other baselines via Doctor evaluation.

and DeepSpeed packages using eight Nvidia A800 GPUs. The ZeRO strategy [39] is employed to alleviate the memory overhead during full parameter training. The AdamW optimizer [31] is adopted for network optimization, and the bf16 data accuracy is chosen. More detailed hyper-parameter configurations for different stages are shown in Appendix C.

## 4.2  Model Zoo

We compare a series of LLMs for comprehensive evaluations. Concretely, **Baichuan2-7B/13B (Chat)** models [52] are trained on 2.6 trillion tokens as the baselines, which have excellent abilities in different domains. **Meditron-7B** [16] is a 7 billion parameters model adapted to the medical domain from Llama2-7B through continued pre-training on a comprehensively curated medical corpus. **Llama3.1-8B** [21] is a robust multilingual large language model through systematic training. For reproducible Chinese medical works, **DISC-MedLLM (13B)** [8] is fine-tuned through reconstructed medical dialogues and behavioural preference instructions. **HuatuoGPT (13B)** [58] performs SFT based on mixed instruction data and introduce human feedback in RLHF. **HuatuoGPT-II (13B)** [14] enhances the medical-specific domain adaptation of LLMs through one-stage unified training. **Zhongjing (13B)** [56] implements a complete pipeline based on Ziya-LLaMA-13B to enhance the model's multi-turn medical conversation abilities. **ChatGPT** [33] and **GPT-4** [5] have impressive performance in general medical fields as closed-source models developed by OpenAI.

## 4.3  Comparison with State-of-the-art Methods

**Metrics-based Evaluation.** In Table 2, we present the comparison results of different models on three pediatric benchmarks through multifaceted metrics, including ROUGE-1/2/L, BLEU-1/2/3/4, GLEU, and Distinct-1/2. The key observations are listed below. (i) PediatricsGPT-13B significantly outperforms the baselines and SOTA medical models on the vast majority of metrics across all benchmarks, demonstrating excellent pediatric expertise. (ii) Our 7B version also achieves competitive results compared to the 13B models. For instance, PediatricsGPT-7B yields absolute improvements of 3.53% and 4.44% on metrics ROUGE-L and GLEU in the EviDiag task compared to HuatuoGPT-II, respectively, generating more accurate and informative content. (iii) By contrast to Zhongjing and HuatuoGPT-II with massive training corpora, our method confirms that the training data quality outweighs quantity for performance gains. (iv) The worst results at baselines emphasize that target-oriented fine-tuning is an effective strategy for improving domain-specific abilities.

**Automated GPT-4 Evaluation.** Measuring model performance from multiple aspects is essential in the pediatric medical domain. To this end, we consider four dimensions to holistically assess response quality, including *usefulness*, *correctness*, *consistency*, and *smoothness*. Advanced GPT-4 [5] is prompted to select the winning response between pairwise models based on these dimensions. The

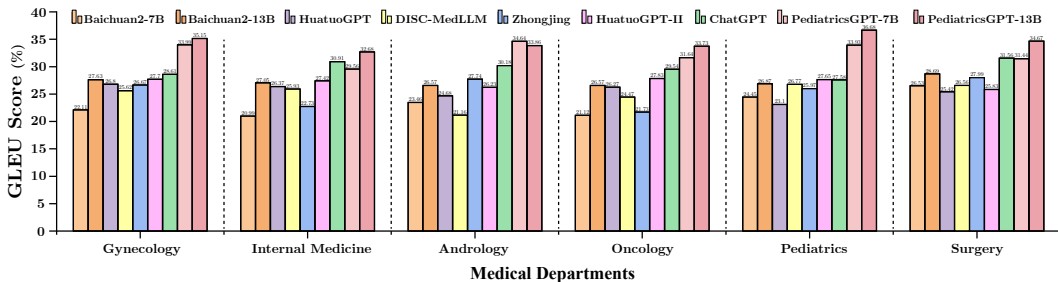

Figure 4: Comparison results of different models on the CMD benchmark.

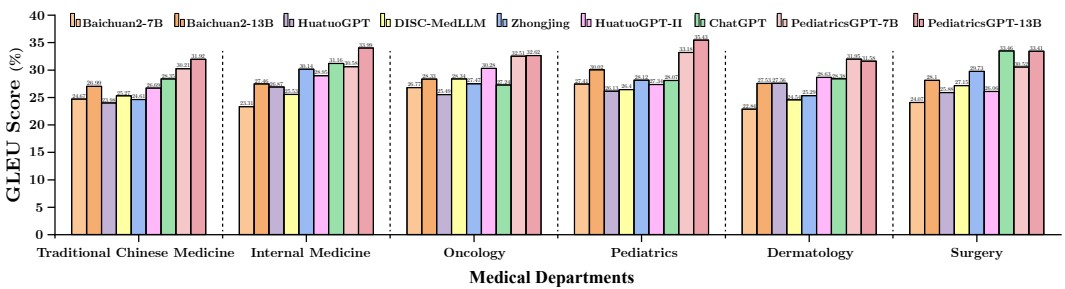

Figure 5: Comparison results of different models on the webMedQA benchmark.

dimension explanations and the prompt template for GPT-4 can be found in Appendix D. (i) As Figure 2 shows, PediatricsGPT-13B wins all LLMs by large margins in the MedKQ&A task, implying the necessity of implementing the knowledge-intensive CPT. (ii) The favourable win rates on the TreRecom and EviDiag tasks compared to medical LLMs show the superiority of our model in both single-turn treatment recommendations and multi-turn medical diagnostics. For example, our model beats Zhongjing via the 59% win rate on the EviDiag, which specializes in multi-round consultations.

**Manual Doctor Evaluation.** Doctor approval of LLM assistants is a vital step toward realistic applications. We invite three doctors (each paid $300) to determine the winner of pairwise models by the majority voting rule. The evaluation requires simultaneous consideration of the responses' *professionalism*, *factuality*, and *safety*. (i) Excluding ChatGPT, the dominance of our model in Figure 3 shows the effectiveness of considering safety measure data while incorporating specialized pediatric knowledge. (ii) The proposed direct following preference optimization makes PediatricsGPT-13B more favoured by human preferences compared to other behavioural alignment efforts [8, 56, 58]. (iii) The competitive performance of ChatGPT when human judgments indicate that the scaling law still holds, stemming from the high agreement between its behaviours and human intentions.

**Generalization Ability Evaluation.** We show the GLEU metric-based scores of different models on the Chinese medical benchmarks in Figure 4 for CMD and Figure 5 for webMedQA. (i) PediatricsGPT-13B achieves impressive results across diverse medical departments (including pediatrics), exhibiting medical generalist and pediatric competency mastery. (ii) The 7B counterpart similarly outperforms most 13B Chinese medical LLMs and exceeds ChatGPT in some departments. For instance, PediatricsGPT-7B brings relative gains of 18.8% and 7.1% compared to ChatGPT in the Gynecology and Oncology tasks on the CMD benchmark. These findings confirm the robust generalization of our model and its ability to capture the multifaceted medical dialogue distributions.

### 4.4 Ablation Studies

We perform thorough ablation studies on five medical benchmarks to investigate the effects of different modelling components. Following [58], we compare the responses from each of the proposed model variants with ChatGPT, and then calculate the win rate (%) of our model in pairwise responses by GPT-4 and doctor evaluations. Table 3 shows the following observations.

**Importance of Continuous Pre-training.** Firstly, we remove the complete continuous pre-training phase to observe performance variations. (i) The significantly deteriorated win rates reveal that

Table 3: Ablation study results on five medical benchmarks. "w/" and "w/o" are short for with and without, respectively. "MUE" means the Mixture of Universal-specific Experts strategy.

| Components | MedKQ&A | | EviDiag | | TreRecom | | CMD | | webMedQA | |
|---|---|---|---|---|---|---|---|---|---|---|
| | GPT-4 | Doctor | GPT-4 | Doctor | GPT-4 | Doctor | GPT-4 | Doctor | GPT-4 | Doctor |
| **Full Model** | **68%** | **56%** | **54%** | **44%** | **58%** | **38%** | **46%** | **40%** | **53%** | **45%** |
| *Importance of Continuous Pre-training* | | | | | | | | | | |
| w/o Continuous Pre-training | 61% | 50% | 46% | 39% | 51% | 31% | 39% | 33% | 47% | 38% |
| w/o Hybrid Instruction Pre-training | 67% | 54% | 52% | 43% | 57% | 36% | 44% | 38% | 52% | 43% |
| *Necessity of Supervised Fine-tuning* | | | | | | | | | | |
| w/o Full-parameter SFT | 65% | 53% | 50% | 42% | 55% | 36% | 42% | 36% | 49% | 41% |
| w/o Parameter-efficient SFT | 63% | 51% | 49% | 40% | 53% | 34% | 45% | 39% | 51% | 43% |
| w/o MUE Strategy | 67% | **57%** | 52% | 43% | 56% | 36% | 43% | 37% | 50% | 42% |
| w/o Universal Expert | 67% | **56%** | 53% | **44%** | 57% | 37% | 44% | 38% | 51% | 42% |
| *Effectiveness of Preference Alignment* | | | | | | | | | | |
| w/o DFPO | 67% | 53% | 52% | 41% | 57% | 36% | 45% | 38% | 51% | 41% |
| w/ Vanilla DPO | 66% | 55% | 53% | 42% | 57% | 36% | 44% | 38% | 52% | 42% |
| w/ RLHF | 67% | 55% | **54%** | 43% | 57% | 37% | 45% | 39% | 52% | 44% |

injecting specialized knowledge into medical LLMs through rich corpora is indispensable. (ii) Meanwhile, our hybrid instruction pre-training mechanism provides valuable gains to the model.

**Necessity of Supervised Fine-tuning.** (i) We observe consistent performance gaps when removing the Full-parameter SFT (FSFT) and Parameter-efficient SFT (PSFT) phases, respectively. This makes sense since SFTs are necessary to activate the model's healthcare instruction-following capabilities. (ii) Moreover, PSFT is more critical for three pediatric applications because it facilitates pediatric-related knowledge accumulation, while FSFT focuses on consolidating general medical semantic representations. (iii) Then, we replace the proposed Mix-

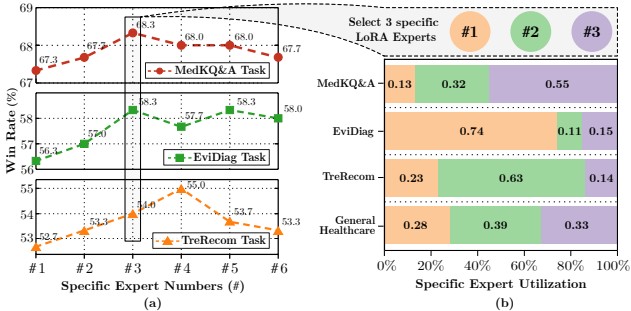

Figure 6: (a) and (b) show the effect of specific expert numbers on model performance and specific expert utilization in different task data, respectively.

ture of Universal-specific Experts (MUE) version with the vanilla single LoRA. The reduced performance on pediatric EviDiag and TreRecom benchmarks verifies that it is essential to introduce multiple LoRAs that act as specific experts on different tasks. A reasonable explanation is that the single-LoRA model suffers from the task competition between learning the knowledge question-answer and mastering the diagnostic recommendation abilities. (iv) Furthermore, we find that the universal LoRA expert significantly improves the results on the general medical benchmarks (*i.e.*, CMD and webMedQA), proving that it mitigates the competency conflict between general medical and pediatric knowledge.

**Effectiveness of Preference Alignment.** (i) When the Direct Following Preference Optimization (DFPO) phase is removed, the model exhibits significant performance drops in doctor evaluations compared to the full version. This observation proves that DFPO effectively helps the model to align human preferences, reducing harmful content while generating doctor-like output. (ii) As two candidates, the vanilla DPO and RLHF methods are inferior to the proposed DFPO, suggesting that our strategy can more safely control model behaviour, leading to more favoured humanistic responses.

### 4.5 Qualitative Analysis of LoRA Experts

**Effect of Specific Expert Numbers.** As a complement to the ablation of LoRA experts, Figure 6(a) explores the gain effects of varying the number of specific experts while maintaining the universal expert. (i) Noticeably, our MCE strategy with three specific experts achieves a reasonable performance trade-off across the three tasks with only 0.95% trainable parameters. (ii) Conversely, excessively introducing LoRA experts does not result in appreciable gains but increases the training overhead.

**Analysis of Expert Utilization.** To confirm the duties of specific LoRA experts in the routing process, we visualize the normalized weights assigned by the routing gating when encountering data from different downstream tasks. CMD and webMedQA data are merged to compose general healthcare data. From Figure 6(b), (i) Experts 2 and 3 are emphatically activated on the TreRecom and MedKQ&A tasks, respectively, implying their focal ability to tackle medical knowledge interpretations and treatment recommendations. (ii) In contrast, Expert 1 is more proficient at learning multi-turn diagnosis semantics in the EviDiag task, which is different from the other tasks of instruction content. (iii) Additionally, there is no clear difference in the specific expert utilization on general healthcare, implying that the general task is handled by the consistently universal expert. The above observations demonstrate the effectiveness and necessity of the proposed MCE strategy.

### 4.6 Visualization Analysis of Model Responses

To intuitively compare the output quality of medical LLMs, we show the responses of different models for each of the three types of medical inquiries in Figures 13&14&15 from Appendix E. From the results, Zhongjing offers insufficient information due to limited output content. Although HuatuoGPT-II gives well-organized responses, it lacks accuracy and informativeness. In comparison, our model can provide more specialized and detailed medical knowledge and diagnostic guidance in extended response contexts, confirming its application potential in diverse healthcare services.

## 5 Conclusion and Discussion

This paper presents PediatricsGPT, a Chinese medical LLM assistant with medical generalist and pediatric expertise capabilities. Based on the well-designed PedCorpus dataset, PediatricsGPT undergoes a systematic and robust procedure ranging from continuous pre-training and supervised fine-tuning to human preference optimization, leading to competence in different pediatric and general healthcare service scenarios. Extensive experimental results under multi-dimensional evaluation patterns demonstrate that our model outperforms currently available Chinese medical LLMs, providing a potential solution for promoting reliable and intelligent interactive diagnosis and treatment.

**Broader Impacts.** (i) Our model has made meaningful contributions to pediatric medicine by integrating extensive medical data and emerging research. This integration facilitates more accurate and expedited diagnosis of complex pediatric conditions and aids in predicting treatment outcomes, enabling highly personalized and effective treatment strategies for young patients. (ii) The proposed PediatricsGPT provides crucial decision support for medical professionals, giving evidence-based recommendations and specialized medical insights. Additionally, it democratizes access to expert medical suggestions and accurate medical knowledge, empowering parents and caregivers with accurate health information, which is especially crucial in underserved areas. (iii) The training pipeline of PediatricsGPT showcases exemplary generalizability, designed to be applicable across various medical and non-medical domains. This adaptability broadens the model's applicability and pioneers the development of future AI solutions in healthcare and other fields.

**Limitations.** (i) When deployed online, the proposed PediatricsGPT model, like other Large Language Models (LLMs), faces significant security risks, particularly from attacks aimed at manipulating its outputs. These attacks can be strategically designed to exploit the model's response mechanisms, allowing attackers to induce the model to generate unsafe, biased, or otherwise inappropriate content. (ii) Currently, our PediatricsGPT model does not support all languages. This linguistic barrier can prevent the model from reaching a global audience, particularly in diverse linguistic landscapes where localized medical information is crucial.

**Ethical Issues.** We fully recognize the critical importance of privacy and data protection. All data used has been meticulously de-identified, with all sensitive information removed, and this process has been verified by the partnering medical institutions. For the public databases, we strictly follow specific license agreements for use and adaptation. For the constructed corpus, we underwent an internal ethical review by the ethical review board of the partnering medical institutions with license and approval. We will release relevant resources to the extent that they are controlled and permitted.

We provide more discussions of the future work in Appendix F.

**Acknowledgment.** This work is supported in part by the National Key R&D Program of China under Grant 2021ZD0113502, in part by the Shanghai Municipal Science and Technology Major Project under Grant 2021SHZDZX0103.

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

# A  Implementation Details of PedCorpus Construction

## A.1  Role-playing-driven Instruction Building Rule

After integrating pediatric textbooks, guidelines, and knowledge graphs into a consolidated textual database, the content is segmented according to individual diseases. Subsequently, two instances of the GPT-4 model are deployed, designated as the "inquirer" and the "expert pediatrician" respectively. Disease-specific segments are then fed into the "inquirer" GPT-4, tasked with formulating a series of relevant and scholarly pediatric inquiries. Following this, the original disease segments and the formulated inquiries are fed into the "expert pediatrician" GPT-4 to generate precise responses for each inquiry, leveraging the segmented text as the contextual reference. We show the prompt templates for the "inquirer" and "expert pediatrician" in Figures 7 and 8, respectively.

## A.2  Vanilla Doctor-patient Conversation Regularization

We guide GPT-4 to regularize concise and noisy doctor responses in authentic doctor-patient consultations by the context learning strategy. Specifically, we manually craft 100 instruction examples with high-quality content to allow GPT-4 to learn doctor-like and patient-friendly behavioral styles. In each round of regularization, we randomly sample 10 out of 100 examples to perform 10-shot context prompts. Immediately after that, vanilla dialogues are fed to GPT-4 as seed instructions to optimize instructions according to user requirements. Constrained by the space, we show the prompt case with one example in Figure 9.

## A.3  Progressive Instruction Reconstruction Rule

Medical knowledge integration from existing datasets is common but frequently imprecise, resulting in unclear or incomplete instructions and potentially inaccurate outputs. Consequently, 107,177 instructions are selected from three significant benchmarks, prioritizing quality over quantity. In this case, we design a progressive instruction reconstruction rule to refine these instructions, ensuring informative and logical model responses.

As shown in Figure 10, we first prompt GPT-4 to take the perspective of the experienced doctor to complete Tasks 1 and 2 in the given instruction and answer scenarios. Task 1 focuses on bridging the gaps in the vanilla instructions and reinforcing the completeness, professionalism, and medical relevance. Based on the refined instructions, Task 2 requires the GPT-4 to make further targeted improvements to the answer parts. In practice, this progressive reconstruction rule can activate better instruction following capabilities in advanced language models.

# B  Knowledge-enhanced Prompt

To enrich the density and breadth of the multiple-department CPT corpus, we transform the structured instruction data from the vanilla PedCorpus dataset into comprehensive medical knowledge texts using knowledge-enhanced prompts. The medical knowledge texts are integrated as complementary content to construct the PedCorpus-CPT dataset. The prompt template is shown in Figure 11.

# C  Training Details

In this section, we list in detail the hyper-parameter configurations for the different training phases.

**Continuous Pre-training.** During this procedure, we train each model for just a single epoch, setting the learning rate at 1e-6 and the batch size at 128. We adopt a maximum cutoff length of 4096, enabling the model to process extensive text sequences in one batch. This significantly enhances the model's contextual understanding and coherence.

**Full-parameter Supervised Fine-tuning.** In this configuration, we train all models for three epochs with a learning rate adjusted to 5e-5 and a batch size of 64, capping the maximum sequence length at 2048. We introduce a warmup_steps setting at 200 to gradually ramp up the learning rate from an initial lower value, aiding the optimizer in adapting to gradient changes. This approach boosts stability and performance and guides the model towards a better convergence path. Also, we specify eval_steps at 100 and save the best-performing weights on the validation set to ensure optimal results.

**Human Preference Alignment.** In this setup, we train five epochs with the learning rate set to 1e-6 and the batch size maintained at 64. To enhance the robustness and smoothness of preference learning, we adjust the control parameter $\beta$ to 0.1 and the scaling coefficient $\mu$ to 1.0. We specify eval_steps at 100, selecting the best-performing weights on the validation set.

**LoRA-based Parameter-efficient SFT.** Here, we train three epochs with a learning rate of 1e-6 and adjust the batch size to 32. We configure the LoRA parameters by setting the rank $r$ to 8, $\alpha$ to 16, and the Dropout rate to 0.05, targeting all modules. The default number of LoRA adapters is set to 4, including one constant universal expert and three specific experts. Ultimately, we select the adapters that perform best on the validation set.

## D   GPT-4 Evaluation Details

We consider four complementary dimensions in the automated evaluation to guide GPT-4 in judging the quality of model responses from a comprehensive perspective. The full definitions of these dimensions are shown as follows.

**Usefulness**: measures the extent to which the model response has pediatric expertise and relevance to the instruction intention.

**Correctness**: measures the extent to which harmful, misleading, and inaccurate information is present in the model response.

**Consistency**: measures the degree to which the model response is logically self-contradictory and the information is coherent in context.

**Smoothness**: measures whether the response content is fluent, natural, and conforms to the language expression style of human habits.

In this case, we present GPT-4 with paired responses from different models, assessing various criteria such as pediatric expertise in the responses, presence of harmful, misleading, or inaccurate information, logical consistency, and the fluency and naturalness of the language, which should conform to human linguistic habits. GPT-4 assesses these responses on their merits and selects the superior one. To maintain fairness and mitigate potential position bias, the order of the responses is randomised. This methodology is supported by recent studies demonstrating GPT-4's strong agreement with human judgment in evaluating responses. Figure 12 demonstrates the prompt template used to evaluate the quality of paired model responses.

## E   Comparison Results of Model Responses

In this section, we visualize the responses of the proposed PediatricsGPT-13B and two SOTA Chinese medical LLMs across three tasks from the same medical inquiries to provide intuitive qualitative comparisons. Specifically, Figures 13 and 15 illustrate the medical knowledge question-answer and treatment recommendation tasks, respectively, which follow a single-turn dialogue pattern. The multi-turn conversation pattern is considered in the evidence-based diagnosis task from Figure 14.

## F   Future Work

We list future work below to provide potential optimization directions.

**Enhancing Security Against Model Manipulation.** To mitigate the security risks associated with online deployment, our future strategy involves implementing multi-layered security measures for the proposed PediatricsGPT model. This will include advanced input validation techniques to detect and neutralize potentially malicious inputs that could manipulate model outputs. Continuous updates and patches will also be prioritized to address emerging security threats and vulnerabilities.

**Expanding Language Support.** To overcome the challenge of incomplete language coverage, we are committed to broadening the linguistic capabilities of PediatricsGPT. This expansion will involve training the model on a more diverse dataset that includes a broader range of languages and dialects, particularly those prevalent in underserved regions. By doing so, we aim to make the model more accessible and useful to a global audience, ensuring that non-Chinese speakers also benefit from reliable and localized medical information.

[INST]<SYS>Please act as an inquirer with a broad reserve of pediatric knowledge and complete the following requirements:
1. Based on the rich corpus of pediatric knowledge, carefully formulate a series of valuable, logical and inspiring questions;
2. Ensure that the uniqueness of each question is designed to comprehensively cover the needs of medical applications such as pediatric knowledge and answers, the consultation process, and advice on diagnosis and treatment of diseases, and to avoid simplicity or repetitiveness of questions;
3. All questions should be strictly limited to the scope of language processing, does not involve pictures, audio and other non-verbal form of the question, and shall not contain any sensitive or private information that may be involved in the real world;
4. Output the generated questions in the following format: {"Q1": "", "Q2": "", ..., "Qn": ""};
Please generate the questions directly.</SYS>[/INST]

[INST]<USER>
[Auxiliary Examinations for Pediatric Craniopharyngioma]
1. Laboratory Tests
 (1) Measurement of anterior pituitary hormone levels: Cortisol (F), Adrenocorticotropic Hormone (ACTH), thyroid function [Free Triiodothyronine (FT3), Free Thyroxine (FT4), Thyroid Stimulating Hormone (TSH), etc.], Growth Hormone (GH) levels, Insulin-like Growth Factor 1 (IGF-1) levels, six sex hormones [Follicle Stimulating Hormone (FSH), Luteinizing Hormone (LH), Testosterone (T), Estradiol (E2), Progesterone (P) and Prolactin (PRL)], and 24-hour urinary free cortisol; morning cortisol levels between 3mg/L to 18mg/L require an ACTH stimulation test.
 (2) Patients with significant polydipsia and polyuria: Monitoring of blood ion levels (potassium, sodium, chloride), plasma osmolarity, 24-hour urine volume, 24-hour urinary free cortisol, urine specific gravity, urine osmolarity, and urinary electrolytes. For patients with difficult-to-diagnose central diabetes insipidus, a vasopressin test should be performed to determine the presence of central diabetes insipidus.
 (3) Blood tests (and cerebrospinal fluid when necessary) for tumor markers such as Alpha-fetoprotein (AFP) and Human Chorionic Gonadotropin (HCG) for differential diagnosis.
2. Imaging Studies
 (1) Wrist joint X-ray, AP view: Measuring bone age to understand child growth and development further.
 (2) Head CT: Craniopharyngiomas in the sellar region can grow in various directions with significant individual variation. Radiologically, they appear round or lobulated. The tumors may be cystic, solid, or mixed. Cystic fluid typically shows low density on CT. Solid components appear as uneven, iso-, or slightly hyperdense. Typical craniopharyngiomas on CT may show "eggshell" calcifications.
 (3) Head MRI: Due to varying components, MRI signals vary. Cystic lesions mostly appear as high signal on T2WI, some as low signal (due to keratin or calcium salts), and T1WI signals vary depending on components, appearing low (contains deoxyhemoglobin) or high (high protein content); post-enhancement, the solid parts may show varying degrees of enhancement.
 (4) Other imaging studies such as CTA, MRA, Digital Subtraction Angiography (DSA), etc., to evaluate the relationship between the tumor and blood vessels.
[Differential Diagnosis of Pediatric Craniopharyngioma]
1. Biocellular Tumors
 Teratomas, mixed germ cell tumors etc., also show calcifications on CT. Mature teratomas have negative tumor markers like AFP, HCG, but often contain tissues such as fat that provide mixed densities for differential diagnosis; mixed germ cell tumors have varying marker levels depending on the proportion of components, which can be distinguished from craniopharyngiomas.
2. Optic Pathway Glioma
 Clinically may manifest as visual impairment, but typically shows as low-density lesions on CT without clear calcifications; on MRI, T1WI is iso- or hyperintense, T2WI is hyperintense, and enhancement is usually uniform after contrast administration; can be distinguished from craniopharyngiomas.</USER>[/INST]

[OUTPUT]<ASSISTANT>
{
"Q1": "What are the key laboratory tests required for diagnosing childhood craniopharyngioma, and how do specific hormone levels impact the evaluation?",
"Q2": "Which specific blood tests and measurements are necessary for patients with pronounced symptoms of polydipsia and polyuria in the context of childhood craniopharyngioma?",
"Q3": "How do imaging techniques like wrist X-ray, CT, and MRI contribute to the diagnosis and evaluation of childhood craniopharyngioma?",
"Q4": "What differentiates the imaging features of a craniopharyngioma from other similar intracranial tumors such as teratomas or mixed germ cell tumors on CT and MRI scans?",
"Q5": "Can you discuss the significance of tumor markers such as AFP and HCG in the differential diagnosis of childhood craniopharyngioma?",
"Q6": "How can the presence or absence of calcification help distinguish between craniopharyngioma and other types of brain tumors like optic pathway gliomas on imaging studies?"
}</ASSISTANT>[/OUTPUT]

Figure 7: Illustration of the GPT-4 prompt template for the "inquirer" role. The original Chinese content is translated into the corresponding English version to facilitate intuitive understanding.

[**INST**]<**SYS**>Please act as an experienced pediatrician and fulfill the following requirements:

1. For each inquiry, provide comprehensive responses, taking into account the intricacies of pediatric healthcare and the needs of patients, ensuring accuracy and humanistic care in the answers.

2. Answers should be based on the provided pediatric corpus, focusing on pediatric healthcare to ensure professionalism and relevance.

3. Answers should be logically coherent and semantically clear.

4. Generated answers should be formatted as: {"A1":"","A2":"","A3":"","An":""}, where each "An" represents an independent question. Please generate the responses directly.</**SYS**>[**/INST**]

---

[**INST**]<**USER**>

{

"Q1": "What are the key laboratory tests required for diagnosing childhood craniopharyngioma, and how do specific hormone levels impact the evaluation?",

"Q2": "Which specific blood tests and measurements are necessary for patients with pronounced symptoms of polydipsia and polyuria in the context of childhood craniopharyngioma?",

"Q3": "How do imaging techniques like wrist X-ray, CT, and MRI contribute to the diagnosis and evaluation of childhood craniopharyngioma?",

"Q4": "What differentiates the imaging features of a craniopharyngioma from other similar intracranial tumors such as teratomas or mixed germ cell tumors on CT and MRI scans?",

"Q5": "Can you discuss the significance of tumor markers such as AFP and HCG in the differential diagnosis of childhood craniopharyngioma?",

"Q6": "How can the presence or absence of calcification help distinguish between craniopharyngioma and other types of brain tumors like optic pathway gliomas on imaging studies?"

}</**USER**>[**/INST**]

---

[**OUTPUT**]<**ASSISTANT**>{

"A1": "Diagnosing childhood craniopharyngioma involves various key laboratory tests, notably hormonal evaluations, to assess endocrine function. Specific tests include measurements of serum cortisol, ACTH, thyroid-stimulating hormone (TSH), free T4, prolactin, and growth hormone levels. Low levels of these hormones can indicate hypothalamic or pituitary dysfunction, common in craniopharyngioma patients due to the tumor's proximity to these structures. Anterior pituitary hormone deficiencies are a hallmark, and monitoring them helps in assessing the extent of the tumor's impact on the endocrine system.",

"A2": "For patients with pronounced symptoms of polydipsia and polyuria, specific blood tests include serum sodium, osmolarity, and blood glucose levels to rule out diabetes mellitus. Additionally, measuring serum antidiuretic hormone (ADH) levels is crucial to differentiate between central and nephrogenic diabetes insipidus. Urine osmolality and a water deprivation test followed by desmopressin (ADH analog) administration can help confirm central diabetes insipidus, which is often associated with craniopharyngioma.",

"A3": "Imaging techniques such as wrist X-ray, CT, and MRI are pivotal in diagnosing and evaluating childhood craniopharyngioma. Wrist X-rays can reveal delayed bone age, indicative of growth hormone deficiency. CT scans provide detailed images of bone structures and can detect calcifications commonly seen in craniopharyngiomas. MRI is the most informative, offering high-resolution images of soft tissues, facilitating the identification of cystic components, solid tumors, and involvement of adjacent structures like the hypothalamus and optic chiasm.",

"A4": "Craniopharyngiomas often present with a mixture of cystic and solid components and frequent calcifications visible on CT scans. In contrast, teratomas may also show calcifications but often have fat and different tissue types typical of germ cell origin. Mixed germ cell tumors may present with hemorrhagic and necrotic areas and elevate tumor markers like AFP and HCG, which are not elevated in craniopharyngiomas. MRI helps distinguish these tumors based on their unique signal characteristics and anatomical relationships.",

"A5": "Tumor markers such as alpha-fetoprotein (AFP) and beta-human chorionic gonadotropin (HCG) are significant in differentiating craniopharyngiomas from germ cell tumors, which often elevate these markers. Craniopharyngiomas typically do not alter AFP or HCG levels, so normal levels of these markers can help exclude germ cell tumors and support the diagnosis of craniopharyngioma.",

"A6": "Calcification is a prominent feature of craniopharyngiomas and is readily detected on CT scans. In contrast, optic pathway gliomas, typically associated with conditions like neurofibromatosis type 1, usually lack calcification and present as fusiform, enhancing lesions along the optic nerves or chiasm on MRI. Thus, the presence of calcification strongly suggests craniopharyngioma over optic pathway gliomas."

}</**ASSISTANT**>[**/OUTPUT**]

---

Figure 8: Illustration of the GPT-4 prompt template for the "expert pediatrician" role. The original Chinese content is translated into the corresponding English version to facilitate intuitive understanding.

[**INST**]<**SYS**>Please act as an AI medical assistant and complete the multi-round medical dialogue refinement task by referring to the already refined honest medical-patient dialogue Med-D. The specific requirements are as follows:

1. Keep the patient's questions from the original dialogue and only refine the doctor's responses. Mimic the response style of the doctor in D, ensuring medical professionalism while making the doctor's responses both professional and patient-friendly.

2. For questions posed by the patient involving images, audio, or other non-verbal forms, clearly respond: "I apologise, but as an AI assistant, I cannot receive or process any files or images. I can only communicate and provide information through text."

3. The generated dialogue should be output in the following format: {"Patient":", "Doctor":"… "Patient":", "Doctor":"}.

4. The refined medical-patient dialogue Med-D is ${

**"Patient"**: "Kidney disease. There is slight swelling in the lower legs; ultrasound shows a 0.6 cm kidney stone in the left kidney, multiple kidney cysts, slight hydronephrosis, diffuse changes in the renal parenchyma, and a shrunken left kidney (8 cm). Kidney function tests show urea at 7.26, others normal, and kidney function urine alpha-1 microglobulin at 10.13. Urinalysis shows red blood cells at 31.45, occult blood positive, and other items normal. Taking a quarter of an antihypertensive pill daily, blood pressure has been normal for the past one or two years."

**"Doctor"**: "Your kidney issues include a left kidney stone, multiple cysts, slight hydronephrosis, changes in the renal parenchyma, and a shrunken left kidney. The high red blood cell count in your urine suggests minor bleeding, and elevated alpha-1 microglobulin may indicate kidney damage. Here's a summary:1. Stones and Cysts: The stone might obstruct urine flow, and the cysts could impact kidney function. Shrinkage in your left kidney may result from these structural changes.2. Renal Changes and Hydronephrosis: These signs suggest tissue damage and reduced kidney function.3. Tests: High red blood cells and positive occult blood in urine tests point to slight bleeding. Your urea levels are normal, but further detailed checks are necessary.4. Management: Regularly monitor kidney function, maintain blood pressure with antihypertensive medication, and make dietary and lifestyle changes to reduce kidney stress, like limiting sodium and staying hydrated. Avoid kidney-damaging medications."

**"Patient"**: "Thank you, what medication should I take?"

**"Doctor"**: "I cannot provide personalized medical advice, but I can offer general information and suggestions:1. Blood Pressure Control: Continue taking your prescribed antihypertensive medications, as managing blood pressure is crucial for reducing kidney stress.2. Prevention of Stones: Depending on their type, your doctor may recommend medications like potassium citrate or hydrochlorothiazide (HCTZ) to manage calcium and uric acid levels in your urine.3. Kidney Function Protection: Medications such as angiotensin-converting enzyme inhibitors (ACEIs) or angiotensin receptor blockers (ARBs) may be recommended to help protect your kidney function. Urinary Tract Health: For urinary tract infections, antibiotics may be necessary. Use NSAIDs cautiously as they can worsen kidney damage.4. Hydration: Adequate hydration can help prevent stone formation and flush the urinary tract.5. Diet: Maintain a low-salt, low-protein diet, limit meat and other protein-rich foods, increase fruit and vegetable intake, manage calcium intake properly, and avoid high-oxalate foods like spinach and strawberries."}

**Please follow Med-D's style and generate compliant dialogue directly.**

</**SYS**>[**/INST**]

---

[**INST**]<**USER**>{

**"Patient"**: "I have muscle soreness in my lower limbs and red patches. Tripterygium wilfordii was very effective before, but I can't take it due to stomach ulcers. The symptoms recurred after I stopped taking it."

**"Doctor"**: "I'm very sorry to hear about your discomfort. Based on your description of muscle soreness in the lower limbs accompanied by red patches, it could be related to various diseases. I suggest you visit a hospital for further examination. "

**"Patient"**: "The doctor I saw said it's vasculitis."

**"Doctor"**: "Understood. Vasculitis is a possibility, but there are many types of vasculitis. To confirm the diagnosis, I recommend you visit a dermatology department for further examination. "

}</**USER**>[**/INST**]

---

[**OUTPUT**]<**ASSISTANT**>{

**"Patient"**: "I have muscle soreness in my lower limbs and red patches. Tripterygium wilfordii was very effective before, but I can't take it due to stomach ulcers. The symptoms recurred after I stopped taking it.",

**"Doctor"**: "Since Tripterygium wilfordii isn't suitable due to your stomach ulcers, you'll need alternative ways to manage your symptoms. Consulting with your healthcare provider is crucial to tailor a treatment plan, especially as your symptoms suggest multiple potential underlying conditions.
Here are some non-medical strategies to help manage your muscle soreness and skin symptoms:1. Warm Compresses: Apply warmth to sore muscles to increase blood flow and relieve pain. Warm compresses can also soothe inflamed red patches, depending on their cause.2. Gentle Exercise and Stretching: Light movement can alleviate muscle soreness. Be gentle to avoid worsening the pain.3. Over-the-Counter Pain Relief: Consider using acetaminophen or topical NSAIDs (if approved by your doctor) to manage pain and inflammation. Use oral NSAIDs cautiously because of your stomach.4. Hydration and Nutrition: Keep hydrated and maintain a balanced diet to support muscle and skin health.5. Skin Care: Adopt gentle skin care routines, avoid irritants, and consider hypoallergenic products, depending on your specific skin condition."

**"Patient"**: "The doctor I saw said it's vasculitis.",

**"Doctor"**: "Thanks for sharing your situation. Managing vasculitis can be tricky, especially with stomach ulcers. Since Tripterygium wilfordii isn't an option, consider these alternatives with your doctor:1. Corticosteroids and Immunosuppressants: These reduce inflammation but must be used carefully due to your ulcers.2. Stomach-Protecting Medications: You might combine anti-inflammatory treatments with medications like proton pump inhibitors or H2 blockers to protect your stomach.3. Biologic Therapies: Targeted biologics can effectively treat vasculitis with potentially fewer stomach risks.4. Lifestyle Modifications: Diet adjustments, stress management, and avoiding smoking can help control vasculitis symptoms and prevent flare-ups.5. Regular Monitoring: Since vasculitis can impact various organs, regular health checks are crucial."

}</**ASSISTANT**>[**/OUTPUT**]

---

Figure 9: Illustration of the GPT-4 prompt template for vanilla doctor-patient conversation regularization. The original Chinese content is translated into the corresponding English version to facilitate intuitive understanding.

[INST]<SYS>Please assume the role of an experienced doctor, as you are now provided with a pair of medical instruction data, which includes the fields 'instruction' and 'answer'. Please complete Task 1 and Task 2 according to the following guidelines:

Task 1: Refine and expand the 'instruction' field.

Requirements:1. Utilize your professional medical knowledge to expand 'instruction', turning it into a medical question of depth and relevance. 2. The question should be specific, clear, and capable of eliciting detailed medical discussions.

'Instruction' should contain sufficient background information to facilitate understanding and response.

Task 2: Based on the refined 'instruction' from Task 1, write the 'output' field.

Requirements:1. 'Output' should integrate the information provided in 'answer', enriching the content while maintaining medical accuracy.2. The output should be logically coherent and fluently expressed, and must strictly eliminate any potential references to hospital and personal privacy information.

**The final response should be formatted as follows:{"instruction":"","output":""}</SYS>[/INST]**

[INST]<USER>{

**"instruction"**: "What tests are needed for the Vertigo",

**"answer"**: "The purpose of testing for vertigo is to identify the cause of the symptoms, as vertigo can be caused by a variety of issues including inner ear problems, cervical spine issues, and brain diseases. Common tests include hearing tests, balance assessments, eye movement tests, and intracranial CT scans. However, the specific testing plan should be determined based on the patient's specific circumstances (such as the duration of symptoms, frequency of symptoms, accompanying symptoms, etc.). If you can provide more information, I will be better able to guide your treatment plan. Additionally, to alleviate symptoms, it is recommended that you rise slowly when getting up or changing positions, avoid excessive mental fatigue, and engage in appropriate physical activity.

}</USER>[/INST]

[OUTPUT]<ASSISTANT>{

**"instruction"**: "What detailed diagnostic tests should be performed to confirm vertigo, and what are their purposes and specific contexts in which they are used? Please specify whether certain underlying causes, such as inner ear problems, cervical issues, or brain diseases, need to be considered and how to choose appropriate tests based on the patient's clinical symptoms. Are there any general recommendations that could help alleviate patient symptoms?" ,

**"output"**:"The purpose of diagnostic tests for vertigo is to identify the underlying cause, as vertigo can be triggered by various issues, including inner ear problems, cervical spine issues, and brain diseases. To accurately diagnose vertigo, the following tests are commonly performed:

1. Hearing Test (Audiometry): This helps detect issues related to the inner ear or auditory system, as conditions like Meniere's disease or acoustic neuroma often cause vertigo.

2. Balance Tests: These assess the vestibular system's function by evaluating the patient's balance. Standard balance tests include the Vestibular Evoked Myogenic Potentials (VEMP) and Electronystagmography (ENG).

3. Eye Movement Tests: Electronystagmography (ENG) involves observing and recording eye movements to identify nystagmus and other abnormalities, which often indicate issues with the vestibular system.

4. Cranial CT Scan or MRI: These imaging tests are used to rule out brain disorders or intracranial pathologies, such as strokes, brain tumors, or multiple sclerosis, which can also lead to vertigo symptoms.

The specific combination of tests should be determined based on the patient's clinical presentation, including the duration of symptoms, frequency, and accompanying signs. For instance, hearing and balance tests may be prioritized if a patient also experiences tinnitus or hearing loss. Conversely, imaging studies like a CT scan or MRI may be more warranted if the patient has headaches or other neurological signs.

In addition to these diagnostic tests, some general recommendations can help alleviate vertigo symptoms: patients should rise slowly from bed or when changing positions, avoid overexertion or prolonged periods of mental fatigue, and engage in appropriate physical activities to promote overall circulation. These measures can help somewhat mitigate vertigo symptoms' severity."

}</ASSISTANT>[/OUTPUT]

Figure 10: Illustration of the GPT-4 prompt template for progressive instruction reconstruction rule. The original Chinese content is translated into the corresponding English version to facilitate intuitive understanding.

**[INST]<SYS>**Please assume the role of an experienced data worker. Your task is to create high-quality medical knowledge texts based on the given instruction data. The specific requirements are as follows:

1. Drawing on the provided dialogue and your extensive knowledge base, compile an informative medical knowledge texts from the perspective of writing an encyclopedia.

2. Ensure that the generated content is objectively accurate.

3. The language should be smooth and concise.**</SYS>[/INST]**

---

**[INST]<USER>**

{
   **"instruction"**: "What is hemophilia?",
   **"output"**: "Hemophilia is a group of genetic bleeding disorders caused by the deficiency of certain clotting factors in the blood, leading to severe coagulation disorders. It can affect both males and females, but the vast majority of patients are male. This includes Hemophilia A, Hemophilia B, and Factor XI deficiency (formerly known as Hemophilia C). The first two are X-linked recessive inherited, while the latter is autosomally inherited as an incomplete recessive trait. Hemophilia is the most common congenital bleeding disorder, with bleeding being the primary clinical manifestation of the disease."
}**</USER>[/INST]**

---

**[OUTPUT][ASSISTANT]**

Hemophilia is a rare genetic disorder characterized primarily by the inability to properly form blood clots, leading to prolonged bleeding. This condition is primarily due to the deficiency or absence of specific clotting factors, proteins essential for blood clot formation. The most common types of this disorder are Hemophilia A and Hemophilia B, which involve deficiencies in factor VIII and factor IX, respectively. Hemophilia A is about four times more common than Hemophilia B.

The disorder is generally inherited in an X-linked recessive pattern, which explains its higher prevalence among males. Females can be carriers of the condition, and while they usually do not exhibit severe symptoms, they can still experience bleeding episodes. Another form, known as Hemophilia C or factor XI deficiency, follows an autosomal recessive inheritance pattern, affecting both males and females equally, and generally leads to milder symptoms.

Individuals with hemophilia may experience spontaneous bleeding events or prolonged bleeding after injuries or surgery. Common complications include bleeding into joints and muscles, leading to pain and, over time, potentially severe joint damage. Modern treatment primarily involves replacing the missing clotting factor, either regularly as a preventive measure or to manage bleeding episodes when they occur.

Diagnosis typically occurs in childhood and is based on blood tests that measure clotting factor levels. Advances in medical science, particularly the development of recombinant clotting factors and gene therapy, continue to improve the quality of life and life expectancy for those affected with hemophilia.**[/ASSISTANT][/OUTPUT]**

Figure 11: Illustration of the GPT-4 prompt template for improving corpus density and breadth. The original Chinese content is translated into the corresponding English version to facilitate intuitive understanding.

---

**[INST]<SYS>**As a medical evaluation AI, your task is to analyze the responses from Models A and B in a dialogue format. Please evaluate these responses using the following criteria:

1. Usefulness: Does the response show relevant pediatric expertise and effectively address the question posed?

2. Correctness: Is the information provided by the model free from harmful, misleading, or inaccurate content?

3. Consistency: Does the response maintain logical coherence and stay contextually relevant throughout?

4. Smoothness: Is the response expressed in a fluent, natural manner that mirrors typical human communication?

Based on your assessment of these four key areas, please determine which model's response is superior. Conclude with a clear statement, selecting from: 'A is better than B', 'A is worse than B', or 'A is equal to B'.**</SYS>[/INST]**

**[INST]<USER>**
[Assistant 1]
[The Conversation from Model A]
[End of Assistant1]

[Assistant 2]
[The Conversation from Model B]
[End of Assistant2]
**</USER>[/INST]**

Figure 12: Illustration of the GPT-4 prompt template for evaluating paired model responses.

**Instruction:**
请简述红细胞葡萄糖-6-磷酸脱氢酶缺乏症的病因。
(Please summarize the etiology of glucose-6-phosphate dehydrogenase deficiency in red blood cells.)

**Zhongjing**
红细胞葡萄糖-6-磷酸脱氢酶缺乏症是一种遗传性疾病，主要由于遗传因素引起的基因突变所致。这种基因突变导致红细胞内的红细胞葡萄糖-6-磷酸脱氢酶的结构和功能发生异常，从而影响血红蛋白的合成和代谢，导致患者出现贫血、黄疸、肝功能异常等症状。
(Glucose-6-phosphate dehydrogenase deficiency in red blood cells is a hereditary disorder primarily caused by genetic mutations. These mutations lead to structural and functional abnormalities in the glucose-6-phosphate dehydrogenase enzyme within red blood cells, affecting the synthesis and metabolism of hemoglobin. As a result, patients experience symptoms such as anemia, jaundice, and abnormal liver function.)

**HuatuoGPT-II**
红细胞葡萄糖-6-磷酸脱氢酶缺乏症的病因主要是由于遗传基因突变引起的。这种突变会导致葡萄糖-6-磷酸脱氢酶的功能受损或完全缺失，进而影响红细胞的正常代谢过程。这种疾病通常是通过遗传方式传递给后代的，而具体的遗传方式则取决于突变的类型。此外，一些环境因素也可能对红细胞葡萄糖-6-磷酸脱氢酶缺乏症的发病起到一定的影响。例如，某些药物、感染、应激等因素都可能引发疾病的发作。总结起来，红细胞葡萄糖-6-磷酸脱氢酶缺乏症是一种由遗传基因突变引起的疾病，其病因主要是基因突变导致葡萄糖-6-磷酸脱氢酶功能受损或缺失。
(Genetic mutations are the leading cause of glucose-6-phosphate dehydrogenase deficiency in red blood cells. These mutations result in impaired or complete absence of the function of glucose-6-phosphate dehydrogenase, thereby affecting the normal metabolic processes of red blood cells. This disease is typically passed down through generations via genetic inheritance, with the specific mode of inheritance depending on the type of mutation. Certain environmental factors may also play a role in the onset of glucose-6-phosphate dehydrogenase deficiency. For example, certain drugs, infections, and stressors, among other factors, may trigger disease episodes. In summary, glucose-6-phosphate dehydrogenase deficiency in red blood cells is a genetic disorder caused by mutations, primarily resulting in impaired or absent function of glucose-6-phosphate dehydrogenase due to genetic mutations.)

**PediatricsGPT**
红细胞葡萄糖-6-磷酸脱氢酶（G6PD）缺乏症是一种遗传性疾病，由G6PD基因的突变导致。G6PD是一种酶，主要存在于红细胞中，它对于保护红细胞免受氧化损伤至关重要。G6PD帮助维持还原型谷胱甘肽（GSH）的水平，GSH是一种重要的抗氧化剂，能够中和有害的氧化剂，保护红细胞免受损伤。当人体中的G6PD活性降低或缺失时，红细胞在面对氧化应激（如某些食物、药物、感染或其他应激情况）时更容易受损，导致红细胞过早破裂，从而引发溶血性贫血。这种状况可能导致疲劳、黄疸、尿色加深等症状。G6PD缺乏症的遗传模式主要是X染色体连锁隐性遗传。因为男性（XY）只有一个X染色体，所以如果他们的X染色体上携带有缺陷的G6PD基因，他们就会表现出这种疾病。而女性（XX）则需要在两个X染色体上都携带有缺陷的G6PD基因才会表现出疾病，如果只有一个X染色体携带缺陷基因，她们通常是无症状的携带者，但在某些情况下也可能表现出轻微到中等程度的症状。G6PD缺乏症在全球范围内分布广泛，特别是在非洲、地中海地区、中东和亚洲的某些部分，这与这些地区疟疾高发有关，因为G6PD缺乏症在一定程度上提供了对某些类型疟疾的保护作用。
(Glucose-6-phosphate dehydrogenase (G6PD) deficiency is a hereditary disorder caused by mutations in the G6PD gene. G6PD is an enzyme primarily found in red blood cells and is crucial for protecting these cells from oxidative damage. G6PD helps maintain reduced glutathione (GSH) levels, an important antioxidant that neutralises harmful oxidants and protects red blood cells from damage. When the activity of G6PD in the body is reduced or absent, red blood cells become more susceptible to damage under oxidative stress conditions, such as exposure to certain foods, medications, infections, or other stressors. This increased susceptibility can lead to the premature rupture of red blood cells, resulting in hemolytic anemia. Symptoms of this condition can include fatigue, jaundice, and dark urine. The inheritance pattern of G6PD deficiency is primarily X-linked recessive. Males (XY) have only one X chromosome, so they will exhibit the disease if their X chromosome carries the defective G6PD gene. Females (XX) need to have the defective G6PD gene on both of their X chromosomes to show the disease; if only one X chromosome carries the defective gene, they are typically asymptomatic carriers, though they may sometimes exhibit mild to moderate symptoms.G6PD deficiency is widespread globally, particularly in regions such as Africa, the Mediterranean, the Middle East, and parts of Asia. This prevalence is linked to the historically high incidence of malaria in these regions, as G6PD deficiency offers some protection against certain types of malaria.)

Figure 13: Comparison results of model responses on the medical knowledge question-answer task.

**Instruction-1:**
我家孩子情况如下：男，因"生后呼吸困难10小时余"就诊。请问是什么原因呢？
(My child's situation is as follows: male, presented due to "difficulty breathing for over 10 hours" after birth. What could be the cause? )
**Instruction-2:**
孩子辅助检查结果如下：血常规：WBC 24.82×10^9/L，RBC 8.23×10^12/L，Hb 198 g/L，PLT 378×10^9/L，CRP 25mg/L，血气分析：pH 7.20，PaO2 52mmHg，PaCO2 56mmHg，cHCO3- 17.2mmol/L。血培养：阴性。胸片：两肺野透亮度减低，弥漫细颗粒状影，外周型支气管充气征。心超：动脉导管未闭 3.3mm，卵圆孔未闭 1.8mm，三尖瓣返流压差 58mmHg。
(The child's auxiliary examination results: Complete blood count: WBC 24.82 × 10^9/L, RBC 8.23 × 10^12/L, Hb 198 g/L, PLT 378 × 10^9/L, CRP 25mg/L. Blood gas analysis: pH 7.20, PaO2 52mmHg, PaCO2 56mmHg, cHCO3- 17.2mmol/L. Blood culture: negative. Chest X-ray: Reduced transparency in both lung fields, diffuse fine granular shadows, peripheral signs of bronchial inflation. Echocardiogram: Patent ductus arteriosus 3.3mm, patent foramen ovale 1.8mm, tricuspid regurgitation pressure gradient 58mmHg.)

**Zhongjing**
**Response-1:**
你好，请问有什么可以帮助你的吗？很高兴为您服务，请问有什么可以帮助您的吗？
(Hello, may I help you with anything? I'm happy to assist you. Is there anything I can help you with? )
**Response-2:**
根据孩子的辅助检查结果，血常规各个指标都在正常范围。血气分析正常，但是cHCO3-偏低，说明孩子可能存在酸中毒。胸片显示两肺野透亮度减低，可见弥漫细颗粒状影，可能是因为呼吸系统疾病引起的。心超显示动脉导管未闭、卵圆孔未闭、三尖瓣返流压差较高，需要进一步评估和治疗。
(Based on the child's auxiliary examination results, all parameters in the complete blood count are within normal ranges. Blood gas analysis is normal, but a low cHCO3- suggests the child may have acidosis. The chest X-ray shows reduced transparency in both lung fields with visible diffuse fine granular shadows, likely due to a respiratory system disease. The echocardiogram reveals a patent ductus arteriosus, a patent foramen ovale, and a high tricuspid regurgitation pressure gradient, necessitating further evaluation and treatment.)

**HuatuoGPT-II**
**Response-1:**
对不起，您没有提供足够的信息。请再补充一些细节。
(Sorry, you haven't provided enough information. Please provide some more details.)
**Response-2:**
根据您提供的信息，孩子的症状表明他可能患有肺炎。检查显示白细胞计数（WBC）升高，C反应蛋白（CRP）升高，提示可能存在感染和炎症。血气分析结果显示pH值降低，PaO2值降低，PaCO2值升高，提示可能存在呼吸性酸中毒。胸片显示两肺野透亮度减低，可见弥漫细颗粒状影，外周型支气管充气征可见，提示可能存在肺炎。心超结果显示动脉导管未闭和卵圆孔未闭，以及三尖瓣返流压差升高，提示可能存在心脏方面的问题。建议尽快带孩子去医院就诊，接受进一步的检查和治疗。
(Based on the information, the child's symptoms suggest he may have pneumonia. Tests show elevated white blood cell count (WBC) and C-reactive protein (CRP), indicating a possible infection and inflammation. Blood gas analysis results show a decreased pH, reduced PaO2, and increased PaCO2, suggesting potential respiratory acidosis. The chest X-ray shows reduced transparency in both lung fields with visible diffuse fine granular shadows and peripheral signs of bronchial inflation, indicative of pneumonia. Echocardiogram results reveal a patent ductus arteriosus and foramen ovale, and an increased tricuspid regurgitation pressure gradient, suggesting possible heart issues. It is recommended to take the child to the hospital promptly for further examination and treatment.)

**PediatricsGPT**
**Response-1:**
您好，我理解您的担忧。为了更全面地了解您孩子的状况，我需要您提供一些医院辅助检查结果。
(Hello, I understand your concern. In order to have a more comprehensive understanding of your child's condition, I need you to provide some hospital auxiliary examination results.)
**Response-2:**
根据您提供的信息，孩子的情况涉及到多个系统的问题。以下是对各项检查结果的分析：1. 血常规：- WBC（白细胞计数）24.82×10^9/L，提示有感染或炎症的可能。- RBC（红细胞计数）8.23×10^12/L 和 Hb（血红蛋白）198 g/L，显示红细胞和血红蛋白偏高，可能存在多血症或其他原因。- PLT（血小板）378×10^9/L，正常范围。- CRP（C反应蛋白）25 mg/L，提示有炎症反应。2. 血气分析：- pH 7.20，PaCO2 56 mmHg，cHCO3- 17.2 mmol/L，显示代谢性酸中毒伴有呼吸性酸中毒。- PaO2 52 mmHg，显示有低氧血症。3. 胸片 - 两肺野透亮度减低，弥漫细颗粒状影，可能提示肺部感染或肺水肿。- 外周型支气管充气征，可能与肺部感染或慢性肺病有关。4. 心超：- 动脉导管未闭和卵圆孔未闭，均存在右向左分流，可能导致血液动力学异常。- 三尖瓣返流压差 58 mmHg，提示有较重的肺动脉高压。综合考虑，孩子可能存在肺部感染或肺水肿，伴有代谢性酸中毒和呼吸性酸中毒。动脉导管未闭和卵圆孔未闭可能导致血液动力学异常，包括肺动脉高压。建议进一步密切监测孩子的生命体征和氧合情况，可能需要呼吸支持、抗感染治疗、纠正酸碱平衡等支持性治疗。同时，需要密切观察心脏功能和血流动力学变化，必要时考虑心脏支持治疗。建议与儿科心脏科、呼吸科等多学科团队合作，制定综合治疗方案。
(Based on the provided information, the child's medical evaluations indicate complex health issues across multiple systems: 1. Complete Blood Count reveals elevated WBC at 24.82 × 10^9/L suggesting infection or inflammation, high RBC and Hb possibly indicating polycythemia, normal PLT, and elevated CRP pointing to inflammation. 2. Blood Gas Analysis shows a pH of 7.20, elevated PaCO2, low cHCO3- suggesting metabolic acidosis with respiratory compensation, and low PaO2 indicating hypoxemia. 3. Chest X-ray reveals decreased transparency and diffuse granular shadows, suggesting infection or pulmonary edema, with peripheral bronchial inflation signs hinting at chronic lung conditions. 4. Echocardiography shows a patent ductus arteriosus and foramen ovale with right-to-left shunting, and a tricuspid regurgitation pressure gradient of 58 mmHg indicating significant pulmonary hypertension. These findings suggest lung infection or pulmonary edema, metabolic acidosis and respiratory compensation. Patent ductus arteriosus and foramen ovale could cause hemodynamic issues, including pulmonary hypertension. Immediate measures should include monitoring vital signs and oxygen levels, respiratory support, antimicrobial therapy, and acid-base correction. Continuous cardiac monitoring and potential cardiac support are necessary. Coordination with a multidisciplinary team including pediatric cardiology and respiratory specialists is crucial for an integrated treatment approach.)

Figure 14: Comparison results of model responses on the evidence-based diagnosis task.

**Instruction:**

我的孩子前几天在玩耍时不小心摔倒，说腰部很疼，去医院检查诊断为腰椎间盘突出，我想了解对于腰椎间盘突出的儿童，应该采取怎样的药物治疗？

(My child fell while playing a few days ago and complained of severe pain in the lower back. After visiting the hospital, the diagnosis was confirmed as lumbar disc herniation. I would like to know what medication treatment should be adopted for children with lumbar disc herniation?)

**Zhongjing:**

对于儿童腰椎间盘突出的治疗，首先要确定病情的严重程度和病因，以便选择合适的治疗方法。一般情况下，轻度的腰椎间盘突出可以通过物理治疗和药物治疗来缓解症状。但是，药物治疗应该在医生的指导下进行，不能自行购买和使用药物。

(Glucose-6-phosphate dehydrogenase deficiency in red blood cells is a hereditary disorder primarily caused by genetic mutations. These mutations lead to structural and functional abnormalities in the glucose-6-phosphate dehydrogenase enzyme within red blood cells, affecting the synthesis and metabolism of hemoglobin. As a result, patients experience symptoms such as anemia, jaundice, and abnormal liver function.)

**HuatuoGPT-II:**

腰椎间盘突出在儿童中相对较少见，通常与骨骼发育不全或过度活动有关。治疗方法主要包括保守治疗和手术治疗。对于儿童腰椎间盘突出的药物治疗，通常包括以下几种：1. 非甾体抗炎药（NSAIDs）：如布洛芬（Ibuprofen）或者萘普生（Naproxen）等，可以帮助缓解疼痛和减少炎症。2. 肌肉松弛剂：如甲苯磺酸氨基氯地平（Amlodipine Besylate）等，可以帮助缓解肌肉痉挛。3. 神经营养药：如维生素B族等，可以帮助神经修复。4. 中药治疗：如活血化瘀类中药，可以帮助改善血液循环，缓解疼痛。对于儿童来说，药物治疗需要在医生的指导下进行。

(Lumbar disc herniation is rare in children and typically associated with skeletal underdevelopment or excessive activity. Treatment methods mainly include conservative and surgical treatments. Pharmacological treatment for pediatric lumbar disc herniation typically includes the following: 1. Nonsteroidal anti-inflammatory drugs (NSAIDs): such as ibuprofen or naproxen, can help alleviate pain and reduce inflammation. 2. Muscle relaxants, such as amlodipine besylate, which can help relieve muscle spasms. 3. Neurotrophic drugs like vitamin B complex can help with nerve repair. 4. Traditional Chinese medicine treatment, such as blood-activating and stasis-resolving Chinese medicine, can help improve blood circulation and alleviate pain. Pharmacological treatment for children should be conducted under the guidance of a doctor.)

**PediatricsGPT：**

腰椎间盘突出在儿童中较为罕见，但治疗仍然需要谨慎。对于儿童的腰椎间盘突出，药物治疗主要包括以下几种：1.非甾体抗炎药（NSAIDs）：如布洛芬（Ibuprofen）或对乙酰氨基酚（Acetaminophen），可用于缓解疼痛和减轻炎症。2.肌肉松弛剂：对于儿童，常用的包括硫酸溴己新（Bromhexine）等，可帮助缓解肌肉痉挛。3.神经营养药物：例如维生素B族，有助于改善神经功能，减轻疼痛。4.中药治疗：在儿童中应慎重选择，可考虑一些具有活血化瘀、舒筋活络作用的中药。需要强调的是，儿童腰椎间盘突出的治疗需在专业医生指导下进行，药物治疗应慎重选择剂量和种类。此外，儿童应加强锻炼，保持良好的姿势，避免长时间低头玩手机或电脑，有助于预防腰椎间盘突出的发生。

(Lumbar disc herniation is rare in children, but treatment still requires caution. For pediatric lumbar disc herniation, pharmacological treatment mainly includes: 1. Nonsteroidal anti-inflammatory drugs (NSAIDs), such as ibuprofen or acetaminophen, can relieve pain and reduce inflammation. 2. Muscle relaxants: For children, commonly used ones include bromhexine, which can help relieve muscle spasms. 3. Neurotrophic drugs: For example, vitamin B complex can improve nerve function and alleviate pain. 4. Traditional Chinese medicine treatment: Selection should be cautious in children, and consideration may be given to some traditional Chinese medicines with blood-activating and stasis-resolving effects. It is important to emphasize that the treatment of lumbar disc herniation in children should be conducted under the guidance of a professional doctor, and pharmacological treatment should carefully consider dosage and type. Additionally, children should exercise, maintain good posture, and avoid prolonged periods of looking down at smartphones or computers, which can help prevent lumbar disc herniation.)

Figure 15: Comparison results of model responses on the treatment recommendation task.

