# OpenReview forum: "PediatricsGPT: Large Language Models as Chinese Medical Assistants for Pediatric Applications"
_NeurIPS.cc/2024/Conference — NeurIPS 2024 poster_

### Official Review · Reviewer_MG71 · 2024-06-23

**Soundness:** 4
**Presentation:** 4
**Contribution:** 3
**Rating:** 7
**Confidence:** 4

**Summary:**

The authors of this paper built PedCorpus, a Chinese pediatric dataset, and PedistrcsGPT, the first Chinese pediatric LLM assistant. Their model was built via continuous pertaining, full-parameter supervised fine-tuning (SFT), direct following preference optimization, and parameter-efficient secondary SFT. Performance evaluation results show that their model outperforms other models.

**Strengths:**

Originality: This is the first time that Chinese LLM has been applied to pediatrics. The author also proposed DFPO, which improves the performance of preference optimization.

Quality and Clarity: Generally, the paper clearly presents the datasets, the methodology, and the performance evaluation, including the experiment settings and results.

Significance: The model could complement the current shortage of healthcare resources in pediatrics.

**Weaknesses:**

1. The techniques in this paper are not that novel, as it applies LLM to another field with main techniques either not original or adapted from others (e.g., DFPO).
2. The paper focuses on the application level. I believe the model could be important if it is deployed. However, there is no evaluation from real users (patients or patients' parents). Doctor's evaluations could be different from the patients' evaluations, especially since this model will be used for children. This weakens this paper's contributions. I think the authors should at least discuss their plan of deployment or intervention as future work.

**Questions:**

1. Do you think DFPO performs better only on your specific task or can generally perform better than RHLF and vanilla DPO? If it can generally perform better, then this can be a great contribution.
2. How will you deploy your model in practice to complement the current shortage of pediatric healthcare resources?

**Limitations:**

The authors list two limitations, which make sense. I wonder whether your model can be 100% accurate. If it generates errors or hallucinations, it should be important to find them and avoid showing them to patients and their parents. As your model will be used for medicine and children, this should be more important than the LLM applied to other fields. This again goes back to my question before: how will you deploy your model in practice?

---

> ### Author Rebuttal · Authors · 2024-08-07
>
> We thank the reviewer for recognizing our new preference optimization and thorough evaluations. We present our detailed responses below.
>
> **Q1**: Clarification of technical novelty.
>
> **A1**: As stated in lines 33-43, this paper addresses the shortcomings at the dataset and framework level of the LLM construction in the Chinese healthcare domain. We propose a high-quality dataset PedCorpus through a series of new instruction-building mechanisms and a systematic training pipeline through new strategies to improve the different challenges in multi-stage training. Our technical novelty is distributed among these mechanisms and strategies. We clarify other contributions besides DFPO mentioned by the reviewer as follows.
> * As stated in lines 54-56, we propose a hybrid instruction pre-training strategy in Continuous Pre-Training (CPT) to bridge the capability weakening due to corpus format discrepancies between the internal and injected medical knowledge of foundation models, facilitating knowledge accumulation and extension.
> * As stated in lines 186-199, we devise a mixture of universal-specific experts strategy to tackle the competency conflict between medical generalist and pediatric expertise in secondary SFT, which strengthens the model's adaptability to distinct downstream tasks. Specific experts master different pediatric tasks by soft routing gating control with noise. Also, a universal expert is consistently activated to prevent general knowledge forgetting and mitigate competency conflict.
> * As stated in lines 106-107, we present a role-playing-driven instruction building mechanism to convert the consolidated textual data into well-designed pediatric instructions. The proposed approach endows the advanced language model with the professional role of an expert pediatrician to generate accurate instruction knowledge for the target model training.
> * As stated in lines 123-124, we design a progressive instruction reconstruction mechanism to distill the sampled instructions to ensure informative pediatric model responses. Unlike the traditional self-instruct pattern of using APIs, our mechanism guides APIs to take the perspective of the experienced pediatrician to complete progressive refinement tasks in the given instruction and answer scenarios.
>
> It is worth noting that these technologies are domain-agnostic and can be absorbed into other communities' LLM constructions to facilitate long-term developments.
> ***
> **Q2**: About the user study and future plan.
>
> **A2**: Constructive proposal. We offer the following two discussions.
>
> **User Study**：With limited rebuttal time and available resources, we did our best to conduct a user study covering 50 patients (each paid $300) in Figure 1 of the **Response.pdf**. Average evaluation results from participants are reported on three benchmarks. We observe similar performance trends as in the original user study from Figure 3, implying the superiority of our model. To investigate the differences between doctor and patient evaluations, we evaluated the consistency of the results from the two user studies using Pearson Correlation Coefficients (PCC). The high PCC score of 0.89 shows that doctor evaluations can reflect patients' preferred outcomes to some extent, suggesting the reasonableness of the evaluation pattern in the main manuscript. The potential reason for this is that patient judgments about personalized medical diagnoses stem largely from interactive knowledge consultations with doctors, resulting in their behaviors being impacted by doctor preferences.
>
> **Future Plan**: We plan to conduct randomized controlled trials with patients from more diverse pediatric medical departments to verify the effectiveness of our model in real-world applications.
>
> We promise to add the above analyses to the revision.
> ***
> **Q3**: Does DFPO perform better on specific or general tasks?
>
> **A3**: In Table 3 of the manuscript, we conducted ablation studies on five benchmarks to investigate the DFPO performance. These benchmarks include three pediatric-specific datasets and two general-purpose datasets with different medical departments. We found that DFPO leads to average gains of 1.4\% and 2.7\% in GPT-4 and doctor evaluations on pediatric tasks, respectively. It can bring average gains of 1.5\% and 2.5\% in GPT-4 and doctor evaluations on general tasks, respectively. In addition, both vanilla DPO and RLHF perform less well than DFPO in the general tasks. These findings confirm that DFPO performs better overall, leading to superior human preference alignment.
> ***
> **Q4**: How to deploy the model in practice?
>
> **A4**: We will adopt two deployment methods.
>
> **Offline Terminal Deployment**: We will deploy smart terminals equipped with our models in our partner healthcare institutions to provide efficient and accurate medical diagnosis and treatment services to patients.
>
> **Online Application Deployment**: We plan to package our modeling system into an application that is easy for users to access on mobile devices. Our application will provide services such as online consultation and consultation record inquiry to reduce the pressure of offline treatment.
> ***
> **Q5**: Regarding the accuracy of the model.
>
> **A5**: We emphasize that no medical LLMs can have 100% accuracy in practice. To avoid errors and hallucinations, we are committed to performing the following measures in our deployments.
>
> * We will combine transformer-based representation editing techniques to manipulate the model to generate harmless responses in the factual semantic space.
> * We will add contrastive decoding strategies to reduce the probability of generating unfaithful content for the next token prediction.
> * We will introduce the retrieval-augmented generation to improve the trustworthiness of medical answers.
> * We plan to build LLM-based agents using external tools for verifiable factual knowledge to ensure response security with real-time interventions.

---

> > ### Comment · Reviewer_MG71 · 2024-08-08
> >
> > Thanks for all your work and clarification. My concerns are addressed.

---

> > > ### Author Response · Authors · 2024-08-08
> > > **Response to Reviewer MG71**
> > >
> > > We thank the reviewer for the meticulous advice!

---

### Official Review · Reviewer_eEmB · 2024-07-12

**Soundness:** 3
**Presentation:** 3
**Contribution:** 3
**Rating:** 6
**Confidence:** 4

**Summary:**

- This paper introduces PediatricsGPT, a Chinese AI assistant for paediatrics
- They created a large dataset (PedCorpus) with 300k+ medical instructions
- The training process is pretty involved - includes pre-training, fine-tuning, and preference alignment
- They came up with some new techniques, like hybrid instruction pre-training
- Evaluation was thorough - used metrics, GPT-4, and even had real doctors test it
- Results show it outperforms other Chinese medical AI models in various health tasks, as well as ChatGPT
- The authors discuss its potential to support doctors and improve pediatric care

**Strengths:**

- It's tackling a real problem - the shortage of pediatric care in China
- The dataset they built (PedCorpus) is very comprehensive and high-quality
- Their evaluation is thorough, using multiple methods including real doctors
- The model outperforms existing Chinese medical AIs, which is impressive

**Weaknesses:**

Overall the paper would benefit from more information on dataset construction, for each specific task and in general. In many cases, it is not clear what is meant by manual sampling, automatic extraction, dataset collection and similar.

It would be great to see the standard deviation for the reported results.

It would be great to see a comparison with human experts, but this is too much to ask for now, so future work.

**Questions:**

Line 12: Immediately, the... Please rephrase, a bit hard to understand

Line 29: Please either reference your own results, or cite a resource that supports your claim.

Line 101: Please add information on how was this done, how were the books selected and how was the knowledge automatically extracted.

Line 109: Given that real patient conversations were used, it would be beneficial to know from where exactly was this data obtained, and were the appropriate ethical guideliens followed.

Line 107: Manually sampled - does this mean a human went through everything and selected the appropriate data.

Line 164-168
- PedCorpus-DFPO is selectively sampled from vanilla PedCorpus - What does it mean "selectively sampled", how was this done? Were there any biases? What parts of PedCorpus were sampled and in which proportion?
- Humanistic stylistic rephrasing - what exactly is this? Who performed the rephrasing (humans or AI, or a combination)? What guidelines were used? How was the quality and consistency of the rephrasing ensured?
- Low-capability medical assistant - please add a bit more info on why was the HuaTuo model chosen, and why is it low capability (how was this verified).

Line 208: How were the 208 difficult tasks sampled, and how was difficulty determined?

The paper would benefit (especially the first couple of pages) from a review of the grammar and sentence clarity.

Note: most of the datasets are in Chinese, maybe some information was obvious, but because of my lack of understanding of the language I was unable to determine that.

**Limitations:**

A very limited validation on real-world use cases, this is more of an initial research paper that can be built upon and further tested in the real world. Would be great to address this in the main part of the paper and not an appendix.

---

> ### Author Rebuttal · Authors · 2024-08-07
>
> We thank the reviewer for recognizing our comprehensive datasets, new techniques, and thorough evaluations. We present our detailed responses below. The common questions regarding the ethical issues can be found in the **global response**.
>
> **Q1**: Explain the dataset construction including textbook selection, automatic extraction, and manual sampling.
>
> **A1**:
> Valuable proposal! We provide detailed explanations based on three parts:
>
> * As stated in lines 101-108, pediatric task-specific data is derived from specialized textbooks, guidelines, and knowledge graphs. In this case, specialized physicians were invited to select valuable textbooks and guidelines from the candidate books. The selection criteria consist of checking the corpus for the presence of informative medical knowledge and up-to-date healthcare content.
> * Suitable natural language processing techniques are used to accomplish the automated extraction process, including named entity recognition (NER) and syntactic parsing. Specifically, we utilize the NLTK tool to extract vital medical terms (e.g., disease names, symptoms, and drug names) from the text. The Stanford Parser-based dependency parsing model is used to analyze sentence structure to find relations among valid medical terms. These techniques select knowledge-intensive passages from the streaming corpus for subsequent instruction data construction.
> * As stated in lines 117-124, the general medical data is mainly from distilled medical datasets. In this case, manual sampling means that we first thoroughly check the completeness of the instructions by considering multiple dimensions, including whether the instructions contain meaningful symptoms and treatments. Instructions are removed if they miss critical information. Next, we employ a BERT-based semantic similarity model to check the instructions information density and select informative parts.
>
> The above procedures help us select appropriate data to construct high-quality PedCorpus. We promise to incorporate the above details in the revision to provide comprehensive insights.
> ***
> **Q2**: Providing the standard deviation of the reported results.
>
> **A2**: Table 4 of the **Response.pdf** provides the mean ± standard deviation of PediatricsGPT results on different benchmarks. We find that the deviations of our model under different metrics across three tasks are within the range of ± 0.8, with most of the deviations of the 13B results in the much smaller interval of ± 0.5. These observations confirm the consistency and effectiveness of the experimental results. We promise to add the above analysis to the revision.
> ***
> **Q3**: Comparison with human experts.
>
> **A3**: We fully agree with the reviewer's proposal because incorporating results from experts helps provide comprehensive insights. To this end, we made the best effort to hire ten pediatricians ($300 each) in limited rebuttals to provide human responses corresponding to test queries on different benchmarks. To better evaluate using ground truth-based metrics, we declared evaluation protocols and required responses to be as well-presented and logical as possible. Table 4 of the **Response.pdf** shows the average results from the human experts. We found that experts performed best on the EviDiag, which requires multi-round consultations. This makes sense since pediatricians have more experience with interactive diagnosis in their practice than the models. The superior performance of our model on knowledge Q&A and treatment recommendation confirms its potential to serve pediatric services.
>
> We are committed to adding expert results to the revision.
> ***
> **Q4**: Description of line 12.
>
> **A4**: Constructive suggestion. We will rewrite the original representation in the revision as "After that, we utilize the full-parameter Supervised Fine-Tuning (SFT) to incorporate the general medical knowledge schema into the models." for intuitive understanding.
> ***
> **Q5**: Reference to line 29.
>
> **A5**: Thanks for the reminder. We promise to reference our own results in the revision to support the stated claim.
> ***
> **Q6**: Related issues of PedCorpus-DFPO.
>
> **A6**:
> * To ensure that the DFPO phase aligns user preferences in different instruction tasks and genres, selective sampling refers to sampling 3889, 3889, 3889, and 3889 instructions from pediatric data, real conversations, Huatuo-26M, and MedDialog, respectively. We extract the original instruction features in these four parts by BERT. Then, K-means clustering is used to identify the instruction cluster features through the optimal 3889 clusters. We select the closest instruction in each cluster as the sampled instruction to avoid data bias.
>
> * Humanistic stylistic rephrasing refers to making the responses as caring and well-presented as physicians. The process is performed by specialized physicians to ensure quality.
>
> * Huatuo (i.e., BenTsao) is chosen to generate low responses because: its performance was the worst on multiple medical tasks in the latest previous study [1]; and we find in practice that its responses were often hallucinatory and inaccurate.
> ***
> **Q7**: Difficult task sampling of line 208.
>
> **A7**: We invited an expert physician from each of the six medical departments to manually sample 50 difficult instances. Difficult tasks were defined by considering two dimensions, including whether they contained rare diseases and whether they required complex logical reasoning.
> ***
> **Q8**: Review of the first couple of pages.
>
> **A8**: Constructive advice. We have reviewed the grammar and sentences of the first couple of pages and promise to fix them in the revision.
> ***
> **Q9**: About real-world use cases.
>
> **A9**: We promise to add more real-world use cases to the main part of the extra page after acceptance.
>
> [1] Yang, S., Zhao, H., Zhu, S. Zhongjing: Enhancing the Chinese medical capabilities of large language model through expert feedback. In AAAI (Vol. 38, No. 17, pp. 19368-19376).

---

> ### Author Response · Authors · 2024-08-10
>
> Dear Reviewer eEmB:
>
> We would like to thank the reviewer for taking the time to review our paper and for the comments.
>
> Please kindly let us know if anything is unclear. We truly appreciate this opportunity to clarify our work and shall be most grateful for any feedback you could give to us.
>
> Best regards,
>
> Authors

---

### Official Review · Reviewer_eZYU · 2024-07-13

**Soundness:** 3
**Presentation:** 2
**Contribution:** 3
**Rating:** 4
**Confidence:** 4

**Summary:**

This paper tried to build an AI-powered pediatric consultation system. Their motivation is current Chinese conversation LLMs for healthcare underperform in pediatric applications due to insufficient instruction data and bad training procedures. To tackle this problem, the authors proposed PedCorpus, a high-quality dataset with over 300K multi-task instructions, and PediatricsGPT, the first pediatric LLM assistant in Chinese. The PediatricsGPT baseline models, enhanced with a hybrid instruction pre-training mechanism, supervised fine-tuning, preference optimization, and a strategy to balance general and pediatric expertise, consistently outperforms previous Chinese conversation LLMs.

**Strengths:**

- The authors are inspired by the new progress in LLMs and then build new datasets for Pediatrics application and new domain-specific LLM model weights for this task.

- The paper's structure is clear and the authors propose a rigorous evaluation pipeline.

**Weaknesses:**

- "Direct Following Preference Optimization (DFPO) in human intention alignment is devised to enhance response robustness and align human preferences." - where is the evidence for this? It would be better to visualize some cases other than using ablation study as we can not see a huge performance difference.

- "ternary instances in the knowledge graphs" How to build the ternary instance for your task.

- "complementary resources" What is a complementary resource?

- Not sure how to protect patient privacy in your Real Doctor-patient Conversations subset.

- As a pediatrician, I don't think the current model has "competence in different pediatric and general healthcare service scenarios". I review your data in the supplementary, it seems most of data do not contain real pediatrics AI challenge. To prove the generalization of the proposed model or prove the value of the dataset in Pediatrics, you should consider building datasets with more specialized pediatric diseases such as autism, cerebral palsy...

- It is also important to test the performance of proposed model in other Out-of-Domain dataset, such as other medical QA dataset for adult.

- It is confusing in your dataset building process. Why choose “expert pediatrician” GPT-4 but not real pediatrician to help you construct the dataset?

- "doctor-like and patient-friendly" - how to define doctor-like and patient-friendly. Do you mean 'professional' here?

- The scale of user study is limited. You state "We invite three doctors (each paid $300) to determine the winner of pairwise models by the majority voting rule." However, it would be better to conduct a wide user study from pediatricians as three people can not represent the human preference of the whole group.

- Institutional Review Board (IRB) Approval is needed as this paper has human user study and the released data includes real doctor-patient conversations.

- Some minor issues:
"Chinese medicine" can be replaced by healthcare service in Chinese as its clinical meaning can be found here: https://www.hopkinsmedicine.org/health/wellness-and-prevention/chinese-medicine

**Questions:**

You can check the Weaknesses part and try to answer the question proposed in each item.

As a physician, I agree this paper has enough novelty and can serve as a useful foundation model in the future. Thus I tend to give Borderline to this paper before reading the rebuttal. However, there are still plenty of issues that need the author to solve them.

During the rebuttal, the author needs to solve the lacking issue of IRB Approval first. If IRB approval is not attached, the dataset and model may contain ethical issues and can not be released.

I am also curious how ChatGPT achieves good medical and pediatric QA performance (only slightly lower than the proposed PediatricsGPT model) with in-context learning.

**Limitations:**

The main limitation is lacking IRB Approval. And most of the data do not contain real pediatrics AI challenges. This dataset can only be considered as a supplement of current released medical QA data for adults in China such as DISC-MedLLM.

---

> ### Author Rebuttal · Authors · 2024-08-07
>
> Insightful comments!  Below are some specific responses. The common questions regarding the ethical issues can be found in the **global response**.
>
> **Q1**: Evidence on the effectiveness of DFPO.
>
> **A1**: We provide evidence both quantitatively and qualitatively.
>
> * Since the degree of preference alignment is difficult to measure with traditional metrics, we evaluated it employing the GPT-4 and Doctor by considering different response dimensions. From Table 3, when DFPO is removed, the model exhibits an average 2.8\% performance degradation across five different benchmarks in Doctor evaluations, specifically a 4\% drop on the webMedQA. These observations prove that DFPO can provide a user-friendly response style that aligns with human preference in expert studies.
>
> * We provide a comparison of the model response with and without DFPO for the same query. We show the English-translated version of the Chinese response in Figure 2 of the **Response.pdf**. The response content is partially omitted due to space constraints. We observe that DFPO helps the model to correct the “no specific treatment” hallucination and provide informative content, leading to better robustness. Moreover, the model response after DFPO is more caring and humanistic, in line with human preferences. We promise to add the qualitative analysis to the revision.
> ***
> **Q2**: How to build ternary instances?
>
> **A2**: The sequential steps are as follows:
>
> * Structured knowledge is extracted from the consolidated textual database using natural language processing methods. Specifically, we used named entity recognition and relationship extraction techniques to identify medical entities (e.g., diseases, symptoms, and medications) and their relationships from the text.
>
> * Then, we assemble ternary instances in the knowledge graph, which contains three parts: subject, predicate, and object.
>
> * We ensure the accuracy of ternary instances through expert validation and manual scrutiny to provide a solid foundation of medical knowledge for our task.
> ***
> **Q3**: What is a complementary resource?
>
> **A3**: Since corpora from textbooks, guidelines, and knowledge graphs have different characteristics, they provide complementary resources for pediatric data extraction. Specifically, textbooks contain authoritative disease content from basic to advanced levels, providing systematic knowledge structures. Guidelines focus on hands-on and diagnostic applications, providing knowledge of specific treatment recommendations. Knowledge graphs contain hierarchical correlation semantics between diseases and symptoms, providing evidence-based diagnostic knowledge.
> ***
> **Q4**: Regarding the model generalization and the dataset value in pediatrics.
>
> **A4**: We want to clarify two points.
> * As shown in Figures 4\&5, we performed extensive comparisons with other baselines on the adult general medical benchmarks CMD and webMedQA, which contain QA data from different departments. These benchmarks belong to the OOD test data since they are not included in the training set. From lines 257-264, our model outperforms previous LLMs in most departments, proving robust generalizability.
>
> * From line 103, PedCorpus incorporates 131 pediatric diseases from 11 broad categories from textbooks to provide specialized pediatric knowledge. From Table 1, PedCorpus also includes data from multiple medical departments to supplement general medical knowledge. In this case, the cases in Supplementary are randomly sampled from the mixed data only to show the instruction construction process. We will open-source pediatric and general medical data separately to provide different values to subsequent studies.
> ***
> **Q5**: Reasons for choosing GPT-4 construct data.
>
> **A5**: We want to clarify that hiring extensive pediatricians working to build large-scale datasets from scratch is labor-intensive and cost-expensive. It is a mainstream and effective strategy to use the advanced GPT-4 that can match human preferences well in a role-playing manner. However, we employed several pediatricians to participate in the instruction checking to ensure the professionalism of the datasets.
> ***
> **Q6**: About doctor-like and patient-friendly.
>
> **A6**: "Doctor-like" means that the response contains professional content like the physician's. "Patient-friendly" means that the response is well-presented, logical, and informative, making it easy for the user to understand.
> ***
> **Q7**: About the user study.
>
> **A7**: With limited rebuttal time, we did our best to conduct a user study covering 50 participants (each paid $300) in Figure 1 of the **Response.pdf**. Average evaluation results from participants are reported on three benchmarks. We observe similar performance trends as in the original user study from Figure 3, implying the superiority of our model. Furthermore, we evaluated the consistency of the results from the two user studies using Pearson Correlation Coefficients (PCC). The PCC score of 0.89 indicates the effectiveness of the small-scale user study. We promise to optimize the related analysis in the revision.
> ***
> **Q8**: About "Chinese medicine".
>
> **A8**: We thank the reviewer for the constructive proposal. We promise to fix it in the revision to avoid the meaning confusion.
> ***
> **Q9**: In-context learning (ICL) for ChatGPT.
>
> **A9**: Insightful comment. We designed a semantics-driven strategy to select suitable examples to guide ChatGPT in improving performance by ICL. During testing, we randomly sample 100 samples from the training instructions to compute the cosine similarity of feature semantics with the test sample. The features are extracted by the BERT model. The training samples with similarity scores ranked top-5 were then used as ICL examples to guide ChatGPT in performing generation. Table 3 in the **Response.pdf** shows the potential of ICL to improve the ChatGPT. However, our model still outperforms ChatGPT on most metrics due to healthcare-specific systematic training.

---

> > ### Author Response · Authors · 2024-08-11
> >
> > Dear Reviewer eZYU:
> >
> > We would like to thank the reviewer for taking the time to review our paper and for the comments.
> >
> > Please kindly let us know if anything is unclear. We truly appreciate this opportunity to clarify our work and shall be most grateful for any feedback you could give to us.
> >
> > Best regards,
> >
> > Authors

---

> > ### Comment · Reviewer_eZYU · 2024-08-12
> > **Thanks for your rebuttal and I will keep my score.**
> >
> > Dear Author of Paper 385,
> >
> > Thank you for your rebuttal and the additional clarifications provided. After reviewing your comments, the rebuttal document, and revisiting the appendix of your original submission, I still have several concerns regarding your paper.
> >
> > The main issue is not with the model architecture itself. I commend you for thoroughly testing various baseline models on your dataset and proposing a new training pipeline. However, it’s important to note that similar solutions have been proposed for general-domain LLMs. Given that your model and data are positioned as the first Chinese pediatric LLM assistant, the primary contribution of this paper should be the quality of the dataset.
> >
> > Your paper makes a strong assertion that LLMs can be effectively used to answer pediatric questions and generate humanistic responses. However, the conversational responses in the dataset provided do not seem to fully support this claim, as they often appear to be direct extracts from textbooks rather than original, contextually adapted answers. Furthermore, it appears that your model, including the dataset, heavily relies on prior work such as DISC-MedLLM. While the concept of developing an LLM specifically for pediatrics is compelling, the current implementation may not adequately address real-world challenges.
> >
> > I suggest consulting with researchers from different pediatric departments before proceeding further with this dataset to ensure that it meets the practical needs of the field.
> >
> > Additionally, we require more time to discuss the ethical implications of your paper with the ethics reviewers once they submit their evaluations. Therefore, I will be maintaining my current score at this time. However, I will make every effort to reach a consensus with the other reviewers in the ML domain, and it is possible that the final score may be adjusted.
> >
> > Best regards,
> > Reviewer eZYU

---

> ### Author Response · Authors · 2024-08-10
>
> Dear Reviewer eZYU:
>
> We would like to thank the reviewer for taking the time to review our paper and for the comments.
>
> Please kindly let us know if anything is unclear. We truly appreciate this opportunity to clarify our work and shall be most grateful for any feedback you could give to us.
>
> Best regards,
>
> Authors

---

> ### Author Response · Authors · 2024-08-11
>
> Dear Reviewer eZYU:
>
> As the discussion period is closing, we sincerely look forward to your feedback. We deeply appreciate your valuable time and efforts. It would be very much appreciated if you could once again help review our responses and let us know if these address or partially address your concerns and if our explanations are heading in the right direction. Please also let us know if there are further questions or comments about this paper. We strive to improve the paper consistently, and it is our pleasure to have your feedback!
>
> Best regards,
>
> Authors

---

> ### Author Response · Authors · 2024-08-11
>
> Dear Reviewer eZYU:
>
> As the discussion period is closing, we sincerely look forward to your feedback. We deeply appreciate your valuable time and efforts. It would be very much appreciated if you could once again help review our responses and let us know if these address or partially address your concerns and if our explanations are heading in the right direction. Please also let us know if there are further questions or comments about this paper. We strive to improve the paper consistently, and it is our pleasure to have your feedback!
>
> Best regards,
>
> Authors

---

> ### Author Response · Authors · 2024-08-13
> **Thanks for the comment and we will make further clarifications about humanistic responses.**
>
> **Q3**：Whether the conversational responses in the dataset support the assertion that the model produces humanistic responses.
>
>
> **A3**: We would like to clarify that Appendix A.1 and Figures 7\&8 show how valid instruction pairs can be extracted from streamed text from specialized pediatric data.  At this stage, we expect the role-play driven API to build instructions with as much guaranteed knowledge completeness and accuracy as possible, rather than contextually adapted answers. Adapted answers will largely suffer from the API's own hallucination problem, leading to potentially noisy and non-factual information. After this stage, we feed the built initial instructions back to the pediatricians to help us with humanistic stylistic rephrasing, referring to making the responses as caring and well-presented as the pediatricians’.
>
> We promise to supplement that the humanistic style comes from the physicians' rewrites after rather than extracting the preliminary instructions in the revision.

---

> ### Author Response · Authors · 2024-08-13
> **Thanks for the comment and we will make further clarifications about similar solutions.**
>
> Dear Reviewer eZYU:
>
> We deeply appreciate the reviewer's comments. Based on your response and original comments, our rebuttal appears to have addressed the vast majority of your concerns. We would like to clarify the remaining concerns as follows.
>
> **Q1**: The reviewer recognizes and appreciates the thorough evaluation and novel training pipeline. However, the reviewer affirms that there are similar solutions in the generalized domain.
>
> **A1**:
> We are concerns that we do not fully understand your insights into similar solutions. It would be helpful if the reviewer could present specific examples for reference that would help explore the interpretation and provide evidence for this statement. Here, we go through the content to help the reviewer address the concern.
>
> We start by listing some of the other reviewers' comments to help you come to a consensus with them, as you hope to do in your response. As described by the **reviewer oou4**, our innovative training process, which includes continuous pre-training, full-parameter supervised fine-tuning, preference optimization, and mixture of universal-specific experts-based supervised fine-tuning, effectively adapts the model to the pediatric domain. The **reviewer jQHJ** explicitly noted the multi-stage training process of this work helped enhance the model's ability to produce good responses and adapt to the complexity of pediatric consultations. In addition, The **reviewers eEmB** and **MG71** pointed out multiple domain-specific innovations in our systematic training pipeline in the strengths, including the new preference optimization strategy DFPO and the hybrid instruction pre-training mechanism. Of course, these new contributions we have also accomplished explicit clarifications in the manuscript.  Specifically,
>
> * As stated in lines 131-138, we propose a hybrid instruction pre-training mechanism in Continuous Pre-Training (CPT) to bridge the capability weakening due to corpus format discrepancies between the internal and injected pediatric knowledge of foundation models, facilitating knowledge accumulation and extension. Our mechanism better facilitates the adaptation of model capabilities in pediatric healthcare.
> * As stated in lines 169-176, we present a stable method for domain-specific LLMs called Direct Following Preference Optimization (DFPO). DFPO utilizes variable changes to formulate the preference loss as a policy function that efficiently optimizes the policy with a simple binary cross-entropy objective. Meanwhile, our method directly regularizes model behavior boundaries in an instruction-following paradigm on medical demonstrations of preferred responses, facilitating robustness and smoothing of the preference learning.
> * As stated in lines 186-199, we devise a mixture of universal-specific experts strategy to tackle the competency conflict between medical generalist and pediatric expertise in secondary SFT, which strengthens the model's adaptability to distinct downstream tasks. Specific experts master different pediatric tasks by soft routing gating control with noise. Also, a universal expert is consistently activated to prevent general knowledge forgetting and mitigate competency conflict. Our strategy enables our models to perform more comprehensively and competitively in pediatric medical applications.
>
>
> We believe that the above contributions can significantly differentiate and strengthen the necessity of our work compared to techniques in the generalized domain.

---

> ### Author Response · Authors · 2024-08-13
> **Thanks for the comment and we will make further clarifications about the primary contribution.**
>
> **Q2**: Given that your model and data are positioned as the first Chinese pediatric LLM assistant, the primary contribution of this paper should be the quality of the dataset.
>
>
> **A2**: As stated in lines 33-43, this paper addresses the shortcomings at the dataset and framework level of the LLM construction in the Chinese healthcare and pediatric domains. We propose a high-quality dataset PedCorpus through a series of new instruction-building mechanisms and a systematic training pipeline through new strategies to improve the different challenges in multi-stage training. Our technical novelty is distributed among these mechanisms and strategies.
>
>
> For the dataset dimension, existing instruction data typically involve vanilla rephrasing of the general medical corpus or aggregation of doctor-like dialogues, which loses the specialization and focus in pediatric applications. More importantly, the current straightforward different round instruction construction paradigms fail to accommodate multi-task healthcare services in real-world scenarios, limiting the model generalization.
>
> In comparison, the novelties of our PedCorpus derive from three characteristics.
> * Task Diversity: besides containing generalist healthcare data, three application-oriented tasks are considered, including medical question-answer, evidence-based diagnosis, and treatment recommendation.
> * Source Richness: distinct pediatric textbooks, guidelines, and knowledge graph resources provide solid assurance of medical knowledge's accuracy.
> * Instruction Extensibility: vanilla instructions can be readily extended to seed instructions for generating specialized corpora to serve different training phases.
>
>
>
> For the technique dimension, prior methods relied on SFT to compensate for medical instruction following capabilities, ignoring the discrepancies between inherent and externally absorbed knowledge within the models. This single pattern causes secondary LLMs to lapse into excessive role-playing rather than understanding. Despite a few attempts in the reinforcement learning from human feedback phases, their performance is restricted by actor-critic instability and online sampling bias.
>
> In addition to the new techniques in the training process mentioned in **A1**, we propose three effective methods in ensuring the quality of the dataset. Specifically,
>
>
> * As stated in lines 106-107, we present a role-playing-driven instruction building mechanism to convert the consolidated textual data into well-designed pediatric instructions. The proposed approach endows the advanced language model with the professional role of an expert pediatrician to generate accurate instruction knowledge for the target model training.
> * As stated in lines 113-155, We introduce a in-context learning paradigm to regularize and refine vanilla conversations through advanced language models with the self-instruct pattern。
> * As stated in lines 123-124, we design a progressive instruction reconstruction mechanism to distill the sampled instructions to ensure informative pediatric model responses. Unlike the traditional self-instruct pattern of using APIs, our mechanism guides APIs to take the perspective of the experienced pediatrician to complete progressive refinement tasks in the given instruction and answer scenarios.
>
>
> Based on the above analysis and explanations, our main contributions are distributed among dataset construction and framework structure, and the different mechanisms and strategies in these two aspects help our work perform better on task-specific pediatric domains and on general purpose medical applications.

---

> ### Author Response · Authors · 2024-08-13
> **Thanks for the comment and we will make further clarifications about  the model and dataset.**
>
> **Q4**:  The model and dataset rely on previous work such as DISC-MedLLM [1].
>
> **A4**: We clarify the differences between our approach and previous work in terms of both the model and the dataset.
>
> **Model Level**: Current Chinese medical LLMs are constructed based on open-source foundation models for secondary development. In this case, we design a systematic training pipeline that incorporates the new direct following preference optimization (DFPO) and the mixture of universal-specific experts (MUE) structures, both of which are absent from previous Chinese healthcare models. Our DFPO facilitates the control of model behavioral boundaries, mitigating potentially harmful and unfaithful outputs. Our MUE structure effectively resolves conflicts between different pediatric tasks and the general medical knowledge, facilitating better knowledge learning and instruction following. In comparison, **DISC-MedLLM [1] does not have extra innovations in model structure and simply uses Baichuan-13B-Base to execute SFT procedures to develop medical competencies**.
>
> **Dataset level**: We elaborate on our PedCorpus  and DISC-Med-SFT from DISC-MedLLM to emphasize that the two are completely different.
>
> As described in Section 3.1\&Table 1 of the manuscript, and the **global response**, the proposed PedCorpus consists of three parts, including pediatric data, real doctor-patient conversations, and distilled medical datasets.
>
> * The pediatric data comes from instruction pairs that we extracted from scratch from specialized textbooks, guidelines, and knowledge graphs, which are key to building models about task-specific pediatric applications.Among these, the knowledge graphs were constructed from scratch by ourselves from the pediatric domain. In contrast, DISC-Med-SFT uses a publicly available knowledge graph, CMeKG, which focuses on general medical knowledge and differs from our data sources and modeling philosophy.
>
>
> * Our doctor-patient conversation data come from two components: voice transcriptions collected at our partnering healthcare institution and the publicly accessible dataset cMedQA2. For the former, we have clarified the relevant ethical issues in the global response and given privacy protection and implementation schemes that match the NeurIPS guidelines. For the latter, we have regularized the vanilla doctor response using a context-learning strategy, which leads to more organized and fluent responses compared to the vanilla instructions. In contrast, DISC-Med-SFT uses the publicly available MedDialog and cMedQA2, which adapts the entire dialogue through prompt engineering. We would like to emphasize that incorporating cMedQA2 in the construction of medical LLMs is a public practice rather than a particular work-specific design. However, the different paradigms for processing vanilla instructions by the different works resulted in completely different final SFT data. These observations and measures imply that doctor-patient conversation data in PedCorpus and DISC-Med-SFT are significantly different.
>
>
> * For other datasets, we found numerous unclear and incomplete representations in the instruction instances from public benchmarks due to the absence of careful calibration, potentially triggering hallucinated outputs. To this end, we sampled a small set of knowledge-intensive instructions following the principle of quality over quantity. We introduced a progressive instruction reconstruction rule to distill the input and output parts of instruction pairs separately, resulting in completely different data compared to vanilla instructions. In contrast, DISC-Med-SFT primarily uses publicly available data from MedMCQA, moss-sft-003, and Alpaca-GPT4.
>
>
> Depending on the citation format we provide, ACs and reviewers can readily access DISC-MedLLM to explore our statements and clarifications.
>
>
>
> [1] Bao, Z., Chen, W., Xiao, S., Ren, K., Wu, J., Zhong, C., ... & Wei, Z. (2023). Disc-medllm: Bridging general large language models and real-world medical consultation. arXiv preprint arXiv:2308.14346.

---

> > ### Comment · Reviewer_eZYU · 2024-08-16
> > **About DISC-MedLLM**
> >
> > Dear Author of Paper 385,
> >
> > I believe there may be some misunderstanding regarding the reason I referenced DISC-MedLLM in my earlier comments. My intention was to highlight that while there are several datasets for medical LLMs with similar formats to yours, the key contribution, as you have stated, should be the quality of the dataset, particularly in the specialized field of Pediatrics.
> >
> > Upon reviewing the data examples provided in both your paper’s appendix and the rebuttal document, I’m concerned that the dataset may not align with what pediatric practitioners would find most valuable for their domain. This is my areas of expertise. DISC-MedLLM encounters similar issues, which is why there’s resistance from healthcare professionals toward over-generalized declarations made by these models.
> >
> > For instance, the data point in your appendix:
> > "My child fell while playing a few days ago and complained of severe pain in the lower back. After visiting the hospital, the diagnosis was confirmed as lumbar disc herniation."
> >
> > The model's response:
> > "Lumbar disc herniation is rare in children, but treatment still requires caution. For pediatric lumbar disc herniation, pharmacological treatment mainly includes: 1. Nonsteroidal anti-inflammatory drugs (NSAIDs), such as ibuprofen or acetaminophen, can relieve pain and reduce inflammation; ... "
> >
> > Similarly, in the rebuttal pdf:
> > "I'm glad to help! Pediatric cerebral palsy is a condition caused by a non-progressive brain injury that affects a child's movement and posture. The goal of treatment is to maximize the child's physical functioning and quality of life. Treatment is usually a multidisciplinary approach that includes the following: 1. Physical therapy: Specific exercises and activities are used to build muscle strength and flexibility and to help the child learn correct posture. 2. Medication: Childhood doses of medications are used to help control symptoms, including anti-spasticity medications such as baclofen, dantrolene, and botulinum toxin injections...... 3. Surgery: Selective posterior spinal rhizotomy or muscle tendon lengthening surgery is required for some children with severe muscle spasticity...... Each child with cerebral palsy is different, and the treatment plan should be individualized and tailored to the child's specific needs and abilities."
> >
> > These responses resemble textbook excerpts or clinical notes, which may not be practical for real-world pediatric practice. This could lead to a significant disconnect between the machine learning and pediatrics communities, something we should aim to avoid.
> >
> > To address this gap, I strongly recommend consulting with pediatric specialists (your clinical collaborators) to review your dataset. Their insights would be invaluable in ensuring that the data reflects the requirements of real-world clinical practice, rather than just verifying the accuracy of the model's reply.
> >
> > Best,
> > Reviewer eZYU

---

> ### Author Response · Authors · 2024-08-13
> **Thanks for the comment and we will make further clarifications about  the practical needs.**
>
> **Q5**: Different pediatric departments were consulted to meet the practical needs of the researcher.
>
>
>
> **A5**: We agree with the reviewer's suggestion. Considering the multifaceted tasks and diagnostic needs in real medical applications, we believe that there is no method currently available that ensures complete generalization to address the challenges in all applications. However, we have the ability to do a good capacity building of models for practical needs on several specific tasks.
> With the suggestions and needs of collaborating healthcare institutions, our work focuses on three applications in pediatric healthcare, including knowledge question-answer, evidence-based diagnosis, and treatment recommendation. Correspondingly, we constructed novel MUE-based SFT processes through task-specific data for learning and mastering relevant knowledge. Also, we incorporates 131 pediatric diseases from 11 broad categories from textbooks to provide the pediatric knowledge needed for these tasks.
>
> From Table 2 and Figures 2\&3, the experiments on the three task-specific testing benchmarks clearly demonstrate the necessity of the model in solving these pediatric tasks. More importantly, the doctor evaluations in Figure 3 also reflect the users' favor and approval of the model responses. Further, we provide more participants in Figure 1 of the **global response** to demonstrate the model effectiveness on more user studies. In addition, we also test the model's generalizability on publicly available healthcare benchmarks webMedQA and CMD with multiple departments, and provide corresponding insights and analyses.
>
> As an exploratory research study, we believe that the current contributions and work are noteworthy and valuable. We are committed to collaborating with more pediatric departments and meeting further needs and tasks in our future plan.

---

> ### Author Response · Authors · 2024-08-13
>
> Dear Reviewer eZYU:
>
> We appreciate the time and effort you have put into reviewing our work. We value the hard-earned time for discussion to address your concerns and make necessary clarifications. We are ready to discuss at any time.
>
> Best regards,
>
> Authors

---

> > ### Comment · Reviewer_eZYU · 2024-08-16
> > **And about the Ethic reviews**
> >
> > I think it is also important to discuss related information about the IRB, dataset released timeline with the ethic reviewers, because the dataset is the biggest contribution of your paper. Can you provide them related documents? And we also need the ACs to encourage the ethic reviewers paticipating the discussion.
> >
> > Best,
> > Reviewer eZYU

---

### Official Review · Reviewer_DSSd · 2024-07-13

**Soundness:** 3
**Presentation:** 3
**Contribution:** 4
**Rating:** 7
**Confidence:** 4

**Summary:**

This paper introduces a Chinese pediatric LLM assistant, PediatricsGPT. It follows a standard pretraining and SFT pipeline to incorporate the general medical knowledge schema into the models. Specifically, they optimize the response to enhance the generation of pediatrician-like humanistic responses.

**Strengths:**

1. The establishment of a large-scale, high-quality multi-task medical instruction dataset, PedCorpus.
2. The development of a Chinese pediatric LLM assistant with pediatric expertise and medical generalist, PedatricsGPT.
3. Open source resources for the community.
4. Extensive experiments on both domain-specific capability and general medical capability.
5. Human (expert) study for a comprehensive evaluation

**Weaknesses:**

1. The authors claim that they crafted 100 high-quality examples to guide the advanced language model, using in-context learning to regularize vanilla conversations in the self-instruct pattern, ensuring doctor-like and patient-friendly model responses. I would like to see a rigorous evaluation of this.

2. According to Table 3, MUE, Universal Expert and RLHF seems to result in a limited performance gain.

**Questions:**

1. When designing such an LLM for pediatrics, is there any specific design/technique regarding such a target group, compared with designing a general medical LLM? I understand the preference optimization for the generation of pediatrician-like humanistic responses is one of such designs; however, is there any other specific design in the training/SFT/knowledge integration phase?

2. In Figures 2&3, how consistent is it for each question in GPT and doctor evaluation?

**Limitations:**

See weakness

---

> ### Author Rebuttal · Authors · 2024-08-07
>
> We thank the reviewer for recognizing our large-scale dataset, open-source resources, and extensive experiments. We present our detailed responses below.
>
> **Q1**: About the evaluation of model responses.
>
> **A1**: Insightful comments! To rigorously evaluate whether the in-context learning (ICL) strategy can ensure that the model produces doctor-like and patient-friendly responses, we did our best to invite 100 patients from different departments at our partnering medical institutions to perform the manual evaluation. Each participant was compensated $300, which was greater than the lowest hourly rate in their region. Manual evaluation is reasonable since the response style measure is not easily quantified by metrics but can be readily evaluated by the users. Constrained by rebuttal time, we chose the 7B size to train a candidate version, called CandidateGPT-7B, without using the ICL strategy to regularize vanilla conversations. Other training procedures and settings were kept consistent for a fair comparison. In this case, we asked participants to consider two dimensions to determine the winning response between PediatricsGPT-7B and CandidateGPT-7B, targeting the same instructions on five benchmarks, including doctor-like professionalism and user-friendliness.
>
> Table 2 in the **Response.pdf** shows the average win/tie/loss rates of PediatricsGPT-7B on different benchmarks across distinct evaluators. We found that PediatricsGPT-7B generated more patient-user-friendly response content with higher win rates on pediatric and general-purpose medical tasks. In addition, the lower tie rates on webMedQA and CMD suggest that the doctor-like and patient-friendly response style of our model can be effectively generalized to single/multi-round consultation scenarios.
> ***
> **Q2**: Performance gains on MUE, universal expert, and RLHF in Table 3.
>
> **A2**: We clarify the gains of different components separately and provide analyses.
>
> * As stated in lines 193-199, in the MUE strategy, specific experts are adaptively activated to resolve knowledge competition across different pediatric tasks. The consistently activated universal expert aims to acquire general medical knowledge to prevent competency conflict between medical generalists and pediatric expertise mastery. When removing MUE in Table 3, we observe drops of 1.5% and 2% in the average results of GPT-4 and Doctor evaluations on the pediatric benchmarks EviDiag and TreRecom, proving that MUE delivers significant gains. The limited gains on MedKQ&A imply that vanilla single LoRA only masters knowledge question-answer semantics, further reflecting the effectiveness of MUE.
>
> * By observing the setting of removing the universal expert in Table 3, we find that the universal expert brings significant performance gains of 2% and 2.5% on average on the general medical benchmarks CMD and webMedQA under the GPT-4 and Doctor evaluations, respectively. In contrast, the expert's limited gains on the three specific pediatric tasks are justified since it is not its duty to acquire task-specific knowledge.
>
> * As stated in lines 169-176, we propose the Direct Following Preference Optimization (DFPO) to solve the unstable reward modeling and high computational costs in RLHF. In this case, the experiments on RLHF in Table 3 belong to the ablation studies that verify the proposed preference optimization approach. The limited gains of RLHF precisely show the merits and necessity of our DFPO in human preference alignment.
> ***
> **Q3**: Other designs/techniques for pediatric LLMs.
>
> **A3**: In addition to the DFPO technique for generating pediatrician-like responses, our study has four other novel designs aimed at the target group.
>
> * As stated in lines 131-138, we propose a hybrid instruction pre-training mechanism in Continuous Pre-Training (CPT) to bridge the capability weakening due to corpus format discrepancies between the internal and injected pediatric knowledge of foundation models, facilitating knowledge accumulation and extension. Our mechanism better facilitates the adaptation of model capabilities in pediatric healthcare.
> * As stated in lines 186-199, we devise a mixture of universal-specific experts strategy to tackle the competency conflict between medical generalist and pediatric expertise in secondary SFT, which strengthens the model's adaptability to distinct downstream tasks. Specific experts master different pediatric tasks by soft routing gating control with noise. Also, a universal expert is consistently activated to prevent general knowledge forgetting and mitigate competency conflict. Our strategy enables our models to perform more comprehensively and competitively in pediatric medical applications.
> * As stated in lines 106-107, we present a role-playing-driven instruction building rule to convert the consolidated textual data into well-designed pediatric instructions. The proposed approach endows the advanced language model with the professional role of an expert pediatrician to generate accurate instruction knowledge for the target model training.
> * As stated in lines 123-124, we design a progressive instruction reconstruction rule to distill the sampled instructions to ensure informative pediatric model responses. Unlike the traditional self-instruct pattern of using APIs, our rule guides APIs to take the perspective of the experienced pediatrician to complete progressive refinement tasks in the given instruction and answer scenarios.
> ***
> **Q4**: Consistency of GPT-4 and Doctor Evaluations.
>
> **A4**: We measure the consistency of the results of two evaluations by Bland-Altman Analysis (BAA) and Pearson Correlation Coefficients (PCC). The mean difference value in BAA is 0.21, and most measurement pairs are within the limits of agreement [-0.78, 1.18], suggesting high agreement between the two evaluations. In addition, the PCC score is 0.93, implying high evaluation consistency. We promise to add the above analyses to the revision.

---

> > ### Comment · Reviewer_DSSd · 2024-08-07
> > **Acknowledgement**
> >
> > Thank you for your response. Most of my concerns have been addressed and I will keep my score.

---

> > > ### Author Response · Authors · 2024-08-08
> > > **Response to Reviewer DSSd**
> > >
> > > Many thanks to the reviewer for the overall constructive and insightful comments!

---

### Official Review · Reviewer_jQHJ · 2024-07-14

**Soundness:** 3
**Presentation:** 3
**Contribution:** 2
**Rating:** 6
**Confidence:** 4

**Summary:**

The paper presents PediatricsGPT, a specialized large language model (LLM) designed to assist in pediatric medical consultations in China, where there is a significant shortage of healthcare resources. PediatricsGPT is built upon PedCorpus, a high-quality dataset containing over 300,000 instructions from pediatric textbooks, guidelines, and knowledge graphs. The model undergoes a systematic training process that includes continuous pre-training, full-parameter supervised fine-tuning, human preference alignment, and parameter-efficient secondary supervised fine-tuning. This comprehensive approach aims to address the inadequacies of existing Chinese medical LLMs in pediatric applications.

**Strengths:**

1) The development of PedCorpus, a robust dataset with over 300,000 multi-task instructions from credible pediatric sources, ensures that PediatricsGPT has access to diverse and accurate medical knowledge.
2) The multi-phase training process, which includes continuous pre-training, full-parameter fine-tuning, and human preference optimization, enhances the model's capability to generate precise responses, and adapt to the complexities of pediatric medical consultations.
3) Experiments show that PediatricsGPT outperforms other Chinese medical LLMs, validating its effectiveness and reliability in real-world applications.
4) Manual human study is conducted.

**Weaknesses:**

1) Overall I believe the developed model with the curated datasets will be a useful resource. But I think the authors can probably better highlight the novelty in this paper and how the techniques employed in this paper differs from other works.
2) As one of the main contribution of this paper lie in the creation of the dataset, I am not sure if this paper is more suitable for the D&B track in NeurIPS. However, this does not affect my overall rating.
3) The improvements attributed to the universal expert appear marginal, raising questions about whether the gains justify the additional time and memory costs.

**Questions:**

See above.

**Limitations:**

Yes

---

> ### Author Rebuttal · Authors · 2024-08-07
>
> We thank the reviewer for recognizing our high-quality dataset, multi-phase training process, and comprehensive experiments. We present our detailed responses below.
>
> **Q1**: Highlight the novelty of this paper and how it differs technically from other works.
>
> **A1**: Thanks for the constructive comments. The novelty of this study encompasses both dataset and methodology technique dimensions.
>
> For the dataset dimension, existing instruction data typically involve vanilla rephrasing of the general medical corpus or aggregation of doctor-like dialogues, which loses the specialization and focus in pediatric applications. More importantly, the current straightforward different round instruction construction paradigms fail to accommodate multi-task healthcare services in real-world scenarios, limiting the model generalization.
>
> In comparison, the novelties of our PedCorpus derive from three characteristics.
> * Task Diversity: besides containing generalist healthcare data, three application-oriented tasks are considered, including medical question-answer, evidence-based diagnosis, and treatment recommendation.
> * Source Richness: distinct pediatric textbooks, guidelines, and knowledge graph resources provide solid assurance of medical knowledge's accuracy.
> * Instruction Extensibility: vanilla instructions can be readily extended to seed instructions for generating specialized corpora to serve different training phases.
>
> For the technique dimension, prior methods relied on Supervised Fine-Tuning (SFT) to compensate for medical instruction following capabilities, ignoring the discrepancies between inherent and externally absorbed knowledge within the models. This single pattern causes secondary LLMs to lapse into excessive role-playing rather than understanding. Despite a few attempts in the reinforcement learning from human feedback phases, their performance is restricted by actor-critic instability and online sampling bias.
>
> In contrast, our technical novelty comes from several points.
>
> * As stated in lines 54-56, we propose a hybrid instruction pre-training mechanism in Continuous Pre-Training (CPT) to bridge the capability weakening due to corpus format discrepancies between the internal and injected medical knowledge of foundation models, facilitating knowledge accumulation and extension.
> * As stated in lines 169-176, we present a stable method for domain-specific LLMs called Direct Following Preference Optimization (DFPO). DFPO utilizes variable changes to formulate the preference loss as a policy function that efficiently optimizes the policy with a simple binary cross-entropy objective. Meanwhile, our method directly regularizes model behavior boundaries in an instruction-following paradigm on medical demonstrations of preferred responses,  facilitating robustness and smoothing of the preference learning.
> * As stated in lines 186-199, we devise a mixture of universal-specific experts strategy to tackle the competency conflict between medical generalist and pediatric expertise in secondary SFT, which strengthens the model's adaptability to distinct downstream tasks. Specific experts master different pediatric tasks by soft routing gating control with noise. Also, a universal expert is consistently activated to prevent general knowledge forgetting and mitigate competency conflict.
> * As stated in lines 106-107, we propose a role-playing-driven instruction building rule to convert the consolidated textual data into instructions. The proposed rule is effective in generating accurate and reliable instruction knowledge for model training.
> * As stated in lines 123-124, we design a progressive instruction reconstruction rule to distill the sampled instructions to ensure informative and logical model responses. Unlike the traditional self-instruct pattern of using APIs, our method guides APIs to take the perspective of the experienced doctor to complete progressive tasks in the given instruction and answer scenarios.
> ***
> **Q2**: About the suitable NeurIPS track for this paper.
>
> **A2**: We thank the reviewer for the overall recognition. As we mentioned in **A1**, this paper presents a high-quality PedCorpus and a systematic training pipeline to address the weaknesses of previous works in both the dataset and modeling methodology. Besides the dataset, our methodological contributions span different stages in the training pipeline to address the corresponding challenges. Specifically, we propose a hybrid instruction pre-training mechanism in the CPT stage to bridge the capability weakening due to corpus format discrepancies between the internal and injected medical knowledge of foundation models. In the intention alignment stage, a DFPO is devised to enhance response robustness and align human preferences. Moreover, we propose a mixture of universal-specific experts strategy to address competition across different pediatric tasks and the conflicts between medical generalization and specialized knowledge in the LoRA-based SFT stage.
>
> We believe that these contributions make this study suitable for the Main Track.
> ***
> **Q3**: Improvements on universal expert.
>
> **A3**: As stated in line 198, the universal expert aims to prevent general knowledge forgetting. In this case, Table 3 clearly shows that the universal expert on the general healthcare benchmark CMD and webMedQA brings significant improvements of 2% and 2.5% on average under the GPT-4 and Doctor evaluation, respectively. Instead, the expert's marginal gains on the three specific pediatric tasks are justified since it is not its duty to acquire task-specific knowledge. Furthermore, the visualization results in Figure 6(b) provide evidence that the universal expert handles general healthcare.
>
> It is worth noting that LoRA-based universal expert requires only 0.24% training parameters, implying almost negligible time and memory cost compared to the whole SFT procedure.
>
> We promise to optimize the above clarifications in the revision.

---

> > ### Comment · Reviewer_jQHJ · 2024-08-11
> >
> > Thank you for the detailed response. I will keep my score.

---

> ### Author Response · Authors · 2024-08-10
>
> Dear Reviewer jQHJ:
>
> We would like to thank the reviewer for taking the time to review our paper and for the comments.
>
> Please kindly let us know if anything is unclear. We truly appreciate this opportunity to clarify our work and shall be most grateful for any feedback you could give to us.
>
> Best regards,
>
> Authors

---

### Official Review · Reviewer_oou4 · 2024-07-20

**Soundness:** 3
**Presentation:** 2
**Contribution:** 3
**Rating:** 5
**Confidence:** 4

**Summary:**

This paper proposes the first Chinese pediatric Large Language Model (LLM) assistant, designed through a robust training pipeline including continuous pre-training, full-parameter supervised fine-tuning, and direct following preference optimization. The model is shown to outperform existing Chinese medical LLMs in pediatric tasks, and the authors intend to open-source both the model and dataset for community development.

**Strengths:**

- First Chinese Pediatric LLM Assistant with Pediatric Expertise: PediatricsGPT represents a significant milestone as the first Chinese LLM specifically designed to assist with pediatric medical applications, demonstrating expertise in this specialized domain.

- Novel Domain Adaptation Paradigm: The paper introduces an innovative training pipeline that includes continuous pre-training, full-parameter supervised fine-tuning, and preference optimization, effectively adapting the model to the pediatric domain.

**Weaknesses:**

- Ethical Issues of the Data: The paper lacks explicit details about the usage agreements for the data sources, including license information and open-source details. Given the data leakage issues inherent in LLMs, clarifying these aspects is crucial for the ethical release and use of both the model and dataset.

- Evaluation Datasets: The three main evaluated datasets appear to be private, raising concerns about their accessibility for further research and verification. The paper should clarify whether these datasets can be openly released and if there are any ethic issues of releasing these datasets.

- Privacy Concerns with GPT-4 API: The use of the GPT-4 API to build instruction data introduces potential privacy concerns, particularly regarding whether data from guidelines and knowledge bases can be legally and ethically fed into a commercialized API like GPT-4.

- Missing Representative Baselines: The paper does not include comparisons with leading medical LLMs such as Meditron, Me-LLaMA, and other advanced models like GPT-4, LLaMA3. Including these baselines would provide a more comprehensive evaluation of PediatricsGPT’s performance.

**Questions:**

N/A

**Limitations:**

Yes

---

> ### Author Rebuttal · Authors · 2024-08-07
>
> We thank the reviewer for recognizing our contribution of significant milestone, novel domain adaptation paradigmm, and innovative training pipeline. Below are some specific responses. The common questions regarding the ethical issues can be found in the **global response**.
>
> **Q1**: About the evaluation datasets.
>
> **A1**: Constructive proposal! As stated in line 21, we are committed to open-sourcing datasets and models to advance the community. In this case, three main evaluated datasets will be made public at the same time to support further research and verification. All components of the database to be released are anonymized and desensitized to eliminate any potential ethical issues. Relevant discussions can be found in the **global response**. We promise to improve the open-source details in the revision.
> ***
> **Q2**: Privacy concerns with GPT-4 API.
>
> **A2**: We want to clarify several aspects to emphasize that the usage of the GPT-4 API in this study does not pose potential privacy concerns.
>
> * The specialized pediatric data used contains only purely medical knowledge from textbooks, guidelines, and knowledge graphs without any personal or sensitive information. Based on this, we further filtered ambiguous and misleading content through meticulous manual screening to avoid any privacy concerns. As stated in lines 106-108 and Appendix A.1, the proposed role-playing-driven instruction construction rule assembles structured instructions only from the consolidated textual data. This means that our rule eliminates privacy concerns in the input stage of the API. In addition, we restrict the generation behavior of the API to a given secure corpus through carefully crafted prompts, which excludes the generation of potentially unethical content from the internal knowledge of the API.
>
> * The collected conversational data has been approved by the institutional ethics committee, which is a prerequisite for us to access the conversational data. All data are strictly anonymized and de-identified to avoid ethical issues. Each participant signed a GDPR informed consent form, which allows the dataset to be publicly available for research purposes. In this case, our in-context learning strategy only regularizes concise vanilla conversations via the API without introducing privacy concerns.
>
> * For the distilled medical datasets, the original gatherers have anonymized the relevant corpus to ensure no personal information is contained. As stated in lines 123-124 and Appendix A.3, our progressive instruction reconstruction rule only enhances the informativeness and density of medical knowledge in the instructions through the API without any privacy leakage.
>
> * The above data resources comply with OpenAI's official terms and security criteria to allow researchers to use the corresponding API in a legal and ethical manner.
>
> ***
> **Q3**: Regarding other representative baselines.
>
> **A3**: Many thanks to the reviewer for the constructive proposal. We clarify the baseline selection and comparison through the following points.
>
> * For a fair and intuitive comparison, the used baselines focus primarily on models that specialize in building Chinese language healthcare capabilities. This means the models' training corpus and response targets are predominantly Chinese rather than English throughout the whole training process.
>
> * We follow the proposal to incorporate the English-oriented baselines to provide comprehensive insights. Among the listed LLMs, we implement models including Meditron-7B-1.0, LLaMA3-8B, and GPT-4. We are considering incorporating Me-LLaMA in the revision as we did not have permission to access its weights before the rebuttal deadline. It is worth noting that the submission deadline (2024/5/22)  for the manuscript was earlier than the release (2024/6/5) of Me-LLaMA's weights.
>
> * Tables 1 in the **Response.pdf** show the comparison results across different tasks on the MedKQ&A, EviDiag, and TreRecom benchmarks, respectively. PediatricsGPT-7B outperforms similarly sized models by large margins, proving the medical expertise of our model. Although other baselines are more competitive on the Distinct metric, which measures response diversity, the precision and recall of responses are more valuable in the medical domain. Furthermore, our 13B version achieves competitive or even better results compared to GPT-4 on the vast majority of metrics, showing the effectiveness of the proposed systematic training pipeline in building domain-specific models.

---

> ### Author Response · Authors · 2024-08-10
>
> Dear Reviewer oou4:
>
> We would like to thank the reviewer for taking the time to review our paper and for the comments.
>
> Please kindly let us know if anything is unclear. We truly appreciate this opportunity to clarify our work and shall be most grateful for any feedback you could give to us.
>
> Best regards,
>
> Authors

---

> ### Author Response · Authors · 2024-08-11
>
> Dear Reviewer oou4:
>
> As the discussion period is closing, we sincerely look forward to your feedback. We deeply appreciate your valuable time and efforts. It would be very much appreciated if you could once again help review our responses and let us know if these address or partially address your concerns and if our explanations are heading in the right direction. Please also let us know if there are further questions or comments about this paper. We strive to improve the paper consistently, and it is our pleasure to have your feedback!
>
> Best regards,
>
> Authors

---

### Author Rebuttal · Authors · 2024-08-07

We thank all reviewers for their time and effort. Here we clarify common questions about ethical issues and open source details of the datasets.

**Q1**: Ethical issues with datasets.

**A1**: As stated in lines 101-124, the proposed PedCorpus consists of three parts, including pediatric data, real doctor-patient conversations, and distilled medical datasets.
* Among these, pediatric data contains only pure medical knowledge from textbooks, guidelines, and knowledge graphs without any private and sensitive information. These resources follow the CC BY 4.0 protocol, which allows users to freely adapt and convert the original data. The creators of these resources are reasonably attributed as co-authors of this study.
* The data source for conversations comes from two components: voice transcriptions collected at our partnering healthcare institution and the publicly accessible dataset cMedQA2 [1]. For the former, we have obtained the approval of the institutional ethics committee, which is a **prerequisite for our access to conversation data**. All data are rigorously anonymized and desensitized to avoid ethical issues. Note that we underwent the internal ethical review by the corresponding Chinese healthcare institution, which is fully compliant with the ethical guidelines of NeurIPS 2024. The corresponding guidelines are described below: **"In cases when no formal process exists, they can undergo an equivalent informal process (e.g., via their peers or an internal ethics review."** Considering the double-blind review policy, we do not have the right to submit the original approval during the review period. If NeurIPS organizers provide special access, we can submit the approval immediately for review. For the cMedQA2, the publisher collected real conversations from online treatment platforms and anonymized personal information. This dataset is licensed under the GPL-3.0 license, which allows users to modify, distribute, and make private use of it.
* In the distilled medical datasets, the protocols for Huatuo-26M, MedDialog, and CMeKG are Apache-2.0, MIT-license, and MIT-license, respectively. These protocols allow users to manipulate the data without restriction, including, but not limited to, the right to use, copy, modify, merge, and distribute. These datasets have been made to protect against privacy leakage. In this case, our instruction reconstruction rule is only used to enhance the density of medical knowledge in the vanilla data.
To summarize, all involved data undergoes rigorous review and processing to comply with ethical guidelines. Our commitment to data privacy and security is unwavering. We promise to refine the usage agreements for the data sources in the revision to provide comprehensive details.

[1] Zhang, Sheng, et al. "Multi-scale attentive interaction networks for chinese medical question answer selection." IEEE Access 6 (2018): 74061-74071.
***
**Q2**: About open source details.

**A2**: As stated in line 21, our database and models will be open source following the MIT license to promote the development of the community. In this case, we promise to release datasets containing training data for model capacity construction and evaluation benchmarks from different tasks and medical departments to avoid researcher concerns about subsequent studies and verifications.
***
**Q3**: Rejection of data leakage in LLMs.

**A3**: This study followed three guidelines to avoid data leakage concerns inherent in LLMs.
* We carefully verified the pre-training details and data usage of the foundation models to exclude any potential leakage of assessment data in the initial model training.
* As indicated in line 206, the data in the designed pediatric medical evaluation benchmarks are derived from held-out samples to reject any overlap with the training data in the systematic pipeline.
* For the general medical proficiency evaluations, we explicitly excluded the used webMedQA and CMD benchmarks from the training data to ensure the fairness and effectiveness of the assessments.

---

> ### Author Response · Authors · 2024-08-16
> **Response to reviewer eZYU**
>
> Dear reviewer eZYU:
>
> As the authors are unable to reply under your comment in openreview at this point in time, we are replying to you in the comments under public rebuttal. We appreciate your time and effort as a professional. We very much agree with your discussion and suggestions on the dataset from your professional perspective as a pediatrician.
>
> As an effort that seeks to explore and develop both in the pediatric and generic domains, we would like to try to build an initial bridge between the machine learning and pediatric communities by providing a new set of training paradigms for large language models as well as initial data benchmarks. Our original intention is to provide some insights into subsequent research, both at the data and modeling level, through our current efforts. The purpose of doing so is to avoid the serious disconnect you mentioned. We believe that someone has to take the first step in order to drive the future of large language model technology to more valuable applications in real-world clinical practice.
>
> We hope that the reviewers will see the valuable and noteworthy contributions of this work, which will be the greatest encouragement for us to continue to work for community development.
>
> Best,
>
> Authors

---

### Comment · Program_Chairs · 2024-08-14

Hi all, the author-reviewer discussion for this paper is extended to Aug 16 11:59pm ET for authors to reply to ethics reviews. -- PCs

---

> ### Author Response · Authors · 2024-08-14
>
> Dear Program Chairs:
>
> Many thanks for your time and effort. We are always ready to discuss for necessary clarifications.
>
> Best regards,
>
> Authors

---

> ### Author Response · Authors · 2024-08-17
>
> Dear Program Chairs:
>
> We looked forward to addressing potential concerns during the extended discussions. However, we have found that the overall discussion seems to be unworkable. We have done our best to respond to comments to avoid misunderstandings and concerns based on multiple responses. We'd like to know if there's anything the authors can do now to.
>
> Best regards,
>
> Authors

---

### Comment · Area_Chair_HJSc · 2024-08-18
**It's reviewer discussion time**

Dear all reviewers,
This is reviewer discussion time. Your peer reviewers' reviews are in, as well as the author responses.
As I understand the process now is when we go in for discussion and aim to form a consensus about this paper.

Can I ask each of you to do the following:
1. Read over each other's reviews
2. Review your own review
3. Look over the author's rebuttal and reviewer's feedback.
4. If you feel like any changes or improvements are needed to your own review -- please update accordingly.
5. Post as a comment to confirm that you've taken a look (just so we know who has/hasn't had a chance to respond yet).

Thanks!

---

### Decision · Program_Chairs · 2024-09-25

**Decision:**

Accept (poster)

**Comment:**

The paper presents PediatricsGPT, the Chinese pediatric Large Language Model assistant, which demonstrates expertise in the specialized domain of pediatrics. The authors propose a novel domain adaptation paradigm and training pipeline, including continuous pre-training, full-parameter supervised fine-tuning, and preference optimization.

The reviewers generally acknowledge the significance of this work as a milestone in Chinese pediatric LLMs and appreciate the innovative training approach. However, several concerns were raised, primarily regarding ethical issues, data privacy, and evaluation methodology.

The main ethical concern was the lack of explicit details about data usage agreements and IRB approval for human subject research. In their rebuttal, the authors clarified that they underwent an internal ethical review by a Chinese healthcare institution, which complies with NeurIPS 2024 guidelines. They also explained that all data sources have appropriate licenses and have been anonymized and desensitized to protect privacy.

Regarding the evaluation datasets, reviewers expressed concerns about their accessibility for further research and verification. The authors addressed this by committing to open-source both the model and datasets under the MIT license, ensuring that the research community can access and verify their work.

The use of the GPT-4 API for instruction data generation raised privacy concerns. The authors provided a detailed explanation of their data handling process, emphasizing that only non-sensitive medical knowledge was used and that strict measures were taken to prevent privacy leakage.

Some reviewers noted the absence of comparisons with leading medical LLMs like Meditron, Me-LLaMA, and GPT-4. The authors acknowledged this limitation and committed to incorporating these baselines in their revised evaluation.

After considering the reviews, rebuttal, and overall contribution of the work, I recommend accepting this paper. The authors have adequately addressed the main concerns raised by reviewers, and the work represents a significant contribution to the field of pediatric LLMs in Chinese. However, I strongly advise the authors to include the promised additional baselines and provide more detailed information about data usage agreements and ethical approvals in the final version of the paper.